# Beyond Self-Interest: How Group Strategies Reshape Content Creation in Recommendation Platforms?

Yaolong Yu [1]   Fan Yao [2]   Sinno Jialin Pan [1]

## Abstract

We employ a game-theoretic framework to study the impact of a specific strategic behavior among creators—group behavior—on recommendation platforms. In this setting, creators within a group collaborate to maximize their collective utility. We show that group behavior has a limited effect on the game's equilibrium when the group size is small. However, when the group size is large, group behavior can significantly alter content distribution and user welfare. Specifically, in a top-$K$ recommendation system with exposure-based rewards, we demonstrate that user welfare can suffer a significant loss due to group strategies, and user welfare does not necessarily increase with larger values of $K$ or more random matching, contrasting sharply with the individual creator case. Furthermore, we investigate user welfare guarantees through the lens of the Price of Anarchy (PoA). In the general case, we establish a negative result on the bound of PoA with exposure rewards, proving that it can be arbitrarily large. We then investigate a user engagement rewarding mechanism, which mitigates the issues caused by large group behavior, showing that PoA $\leq K+1$ in the general case and PoA $\leq 2$ in the binary case. Empirical results from simulations further support the effectiveness of the user engagement rewarding mechanism.

## 1. Introduction

With the increasing popularity of online media, content creation has emerged as a prominent profession in contemporary society (Bobadilla et al., 2013; Hoose & Rosenbohm, 2024). Driven by platforms' rewarding mechanisms, creators can generate income proportional to their output and often strive to maximize their earnings by optimizing their content, such as selecting formats or topics that yield higher profitability (Glotfelter, 2019; Hodgson, 2021). To maximize user utility, leading platforms like Instagram, TikTok, and YouTube employ sophisticated algorithms to match users with the most relevant content. This process involves calculating relevance scores and applying top-$K$ ranking methods to prioritize and display content that aligns closely with user preferences.

However, recent studies have shown that user utility cannot be maximized solely by optimizing matching rules, as creators can act strategically, leading to dynamic shifts in content distribution (Ben-Porat & Tennenholtz, 2018; Yao et al., 2024c;b; Jagadeesan et al., 2023; Hron et al., 2022). Previous works mainly study this problem under self-played content creators' strategic behavior, where every content creator only focuses on maximizing its utility. As online recommendation platforms become extremely prevalent these days, despite individual creators, some creators form a group to strategically act in this content creation competition, usually united by a media company, such as an MCN (Gardner & Lehnert, 2016; Gardner, 2015; Boyle & Boyle, 2018; Liang & Ji, 2024). To this group of creators, the company needs to consider assigning and adjusting topics for each creator in this group to maximize the group's utility.

Within a group of creators, competition among members can be strategically reduced to enhance collective utility. Importantly, the strategies that maximize group utility often differ significantly from those that maximize individual utilities when creators act in a self-interested manner. The implications of group behavior extend beyond merely improving group utility: such strategic collaboration can alter the equilibrium of the content creation competition, resulting in a content distribution that diverges from the equilibrium observed when only individual creators are present. This shift in content distribution can influence user satisfaction with recommended content and may potentially diminish overall user welfare. For instance, consider a scenario where the user base consists predominantly of sports fans, with only a small fraction interested in niche topics. When creators collaborate to boost revenue, they may shift from sports

---

[1]Department of Computer Science and Engineering, The Chinese University of Hong Kong, Hong Kong, China [2]Department of Computer Science, University of Virginia, Charlottesville, Virginia, USA. Correspondence to: Sinno Jialin Pan <sinnopan@cuhk.edu.hk>.

*Proceedings of the 42$^{nd}$ International Conference on Machine Learning*, Vancouver, Canada. PMLR 267, 2025. Copyright 2025 by the author(s).

to niche topics to minimize internal competition. Consequently, the limited supply of sports-related content may fail to meet the demand of the majority sports fan population. Thus, it is crucial to analyze the impact of this emerging group behavior on content distribution and user welfare, as it holds significant implications for the design and regulation of recommendation systems. In this study, we analyze the problem within a well-defined game-theoretic framework—the Content Creation Competition ($C^3$) introduced by Yao et al. (2023; 2024a;c)—under a top-$K$ recommendation mechanism with exposure reward, which is widely used by online content platforms (Ben-Porat & Tennenholtz, 2018; Hron et al., 2022; Jagadeesan et al., 2023; Meta, 2022; Savy, 2019; Eilat & Rosenfeld, 2023). We begin by examining a simplified scenario, termed the bandit $C^3$ game, where users' features and creators' action sets are represented as orthogonal unit basis vectors in $\mathbb{R}^n$. We define the group equilibria and establish constraints under which the group equilibrium remains consistent with the individual case. While these constraints ensure that group behavior does not alter the content distribution, they are neither generally applicable nor practical in most real-world settings. We then investigate the resulting user welfare loss when these constraints are relaxed.

We show that, due to the zero-sum like structure of the exposure-based rewarding mechanism, creators within a group can reduce internal competition to improve group utility. This strategic behavior can potentially harm user welfare. The worst-case scenario occurs when all creators form a single group, leading to a user welfare loss as large as $O(n)$.

We also show that the parameters $K$ and $\beta$ in the exposure rewarding mechanism can significantly influence user welfare. First, we demonstrate that in the case where users pay constant attention to the recommended items, increasing competition—such as by increasing $K$ or $\beta$—can improve user welfare. However, given the diminishing attention spans of users (Carr, 2020), we present contrasting results. As highlighted in prior work (Yao et al., 2023), a key insight in top-$K$ mechanisms is that user welfare tends to improve as $K$ or $\beta$ increases, as both parameters augment the expected number of creators exposed to users (Yao et al., 2024c). This is intuitive, as users are more likely to find satisfactory content when presented with more choices. However, we present a counterintuitive finding in the context of group behavior and the diminishing attention spans of users: user welfare does not necessarily increase with larger values of $K$ or $\beta$. Together, these findings provide insights for selecting $K$ and $\beta$ in different practical scenarios, particularly when considering group dynamics.

In more general scenarios, we first analyze the worst case for a $C^3$ game with exposure reward, showing that the Price

of Anarchy (PoA) can be arbitrarily bad. This is mainly due to the inefficiency of the zero-sum structure in motivating creators to generate high-quality content when certain groups dominate the entire exposure of users. We then investigate another rewarding mechanism — user engagement rewards, which are deemed more effective for maximizing welfare as they better align with user preferences (Acharya et al., 2024; Yao et al., 2023). Engagement rewards alleviate the worst-case issue in exposure rewards, motivating creator groups to produce high-quality content and enhance user welfare. We establish an upper bound on the PoA for the $C^3$ game with user engagement-based rewards and group behavior, showing that the PoA can be as large as $K + 1$. To complement this, we construct a worst-case instance and derive a lower bound, demonstrating that the lower bound nearly matches the upper bound. Although this PoA result is negative, the worst-case scenario can be effectively mitigated through intervention methods (Yao et al., 2024c). Additionally, we demonstrate positive results in specific cases—such as when relevance scores and user attention scores are binary—showing that the PoA of this game is no more than 2. We construct simulations to further validate the effectiveness of the user engagement rewarding mechanism.

## 2. Related Work

The study of game-theoretic aspects in recommendation systems began with the seminal works of Ben-Porat & Tennenholtz (2018; 2017), which introduced a recommendation system based on the Shapley value. This approach guarantees the existence of a unique PNE while satisfying several fairness criteria. Recently, numerous studies have delved into the dynamics of strategic content creators, a topic that has attracted significant attention in both theoretical and empirical research (Yao et al., 2023; 2024a;c; Zhu et al., 2023; Hu et al., 2023; Hron et al., 2022; Dean et al., 2024b;a; Mladenov et al., 2020; Prasad et al., 2023). This growing body of work underscores the importance of understanding creators' strategic behaviors and platform design to achieve sustainable and equitable outcomes in the long term. A common assumption in these studies is that creators are self-interested, with their strategic behavior analyzed under the mediation of a recommendation system. In contrast, our work diverges from this assumption by considering creators who form groups and strategically optimize their collective utility.

Hron et al. (2022) studied equilibria in exposure games, where creators are incentivized by exposure-based rewards. Concurrently, Jagadeesan et al. (2023) explored a related problem, focusing on supply-side competition in scenarios where creators operate within a high-dimensional strategy space. Their study focused on characterizing NE and identifying conditions under which specialization in creators'

strategies may emerge. Yao et al. (2023; 2024a) studied creator dynamics within the $C^3$ framework. Building on this framework, we analyze creators' strategic group behavior and its implications for user welfare. Recent work (Yao et al., 2023) highlights the impact of content creators' competitive behavior on user welfare in top-$K$ recommendation systems, suggesting that the user welfare guarantee improves as either $K$ increases or $\beta$ increases. However, in our work, we demonstrate that user welfare no longer necessarily increases with larger values of $K$ or temperature $\beta$ when considering the group behavior of creators.

To maximize long-term user welfare, Hu et al. (2023) developed a learning algorithm for the platform to encourage the production of high-quality content, Zhu et al. (2023) proposed an online learning approach that optimizes the recommendation policy and payment contracts for creators, and Yao et al. (2024c) provided intervention mechanisms for the platform to steer the equilibrium content distribution towards a desirable user welfare outcome. Specifically, Yao et al. (2024c) demonstrates that the platform can signal the importance of a specific user $\boldsymbol{x}$ by increasing $K$ or $\beta$, which enhances $\boldsymbol{x}$'s visibility among creators. This increases the chance that creators who were previously unaware of $\boldsymbol{x}$ recognize the potential benefits of catering to $\boldsymbol{x}$. We also analyze the roles of these two parameters in our setting, showing that larger values of $K$ or $\beta$ introduce additional competition to the $C^3$ game, leading to distinct content distribution and user welfare, which provides insights that the platform can improve user welfare by appropriately tuning $K$ and $\beta$ under different user attention scores on content.

## 3. Preliminary

In this section, we formalize the game-theoretic framework—the Content Creator Competition ($C^3$) game—with the platform's rewarding mechanisms. This strategic game setup builds upon the framework introduced by Yao et al. (2023; 2024a). Each instance of the $C^3$ game, denoted by $\mathcal{G}$, is described by a tuple $(\{\mathcal{S}_i\}_{i=1}^n, \mathcal{X}, \sigma, \{r_i\}_{i=1}^n, \{\mathcal{C}_\ell\}_{\ell=1}^L, \beta, K)$, as illustrated below:

1. **Basic setups:** a user distribution $\mathcal{X}$ with finite support $\{\boldsymbol{x}_j \in \mathbb{R}^d\}_{j=1}^m$, where $\boldsymbol{x} = \boldsymbol{x}_j$ with probability $\{p_j\}_{j=1}^m$, and a set of content creators denoted by $[n] = \{1, \cdots, n\}$. Each creator $i$ can take an action $\boldsymbol{s}_i$, referred to as a *pure strategy* in game-theoretic literature, from an action set $\mathcal{S}_i \subset \mathbb{R}^d$. Here, $\boldsymbol{s}_i$ is the embedding of content creator $i$ will produce. Without loss of generality, $\|\boldsymbol{x}\|_2 \le 1, \|\boldsymbol{s}_i\|_2 \le 1$.

2. **Relevance function:** the relevance function $\sigma(\boldsymbol{s}, \boldsymbol{x}) : \mathbb{R}^d \times \mathbb{R}^d \to \mathbb{R}_{\ge 0}$ measures the *relevance* score between a user $\boldsymbol{x} \sim \mathcal{X}$ and content $\boldsymbol{s}$. Without loss of generality, we normalize $\sigma$ to $[0, 1]$, where 1 suggests perfect

matching. We focus on modeling the strategic behavior of creators and thus abstract away the estimation of $\sigma$.

3. **Matching probability:** Given any user $\boldsymbol{x} \in \mathcal{X}$ and when each creator commits to a strategy $\boldsymbol{s}_i$, the platform retrieves the top-$K$ ranked content in terms of the relevance scores $\{\sigma(\boldsymbol{s}_i, \boldsymbol{x})\}_{i=1}^n$. Specifically, let $\{\sigma_{l(1),\boldsymbol{x}} \ge \cdots \ge \sigma_{l(n),\boldsymbol{x}}\}$ be a permutation of $\{\sigma(\boldsymbol{s}_i, \boldsymbol{x})\}_{i=1}^n$. The matching probability of $\boldsymbol{s}$ to user $\boldsymbol{x}$ is calculated using a softmax distribution with temperature $\beta > 0$, i.e., for $1 \le l(i) \le K$,

$$P_i(\boldsymbol{s}, \boldsymbol{x}) \triangleq \mathrm{Prob}[\boldsymbol{s} = \boldsymbol{s}_{l(i)}] \propto \exp[\beta^{-1}\sigma_{l(i),\boldsymbol{x}}].^1 \quad (1)$$

A small $\beta$ makes the matching strategy more deterministic, and $\beta \to \infty$ corresponds to random matching.

4. **User utility and user Welfare definitions**: We define user $\boldsymbol{x}$'s utility from consuming a list of ranked content as follows:

$$\pi(\boldsymbol{s}, \boldsymbol{x}) = \sum_{i=1}^n r_i \sigma_{l(i)}(\boldsymbol{x}) \cdot \mathbb{I}\{i \le K\},$$

where $1 \ge r_1 \ge r_2 \ge \cdots \ge r_n \ge 0$. The sequence $\{r_i \in [0, 1]\}_{i=1}^n$ represents the user's attention scores over the $k$-th ranked content, where content with a higher relevance score receives more attention. The user welfare $W(\boldsymbol{s})$ is defined as the expected user utility:

$$W(\boldsymbol{s}) = \mathbb{E}_{\boldsymbol{x} \sim \mathcal{X}}[\pi(\boldsymbol{s}, \boldsymbol{x})]. \quad (2)$$

5. **Creator utility**: Exposure reward:

$$u_i(\boldsymbol{s}) = \mathbb{E}_{\boldsymbol{x} \in \mathcal{X}}[\mathbb{I}\{\sigma_i(\boldsymbol{s}, \boldsymbol{x}) > 0\}P_i(\boldsymbol{s}, \boldsymbol{x})]. \quad (3)$$

This exposure rewarding mechanism is widely used in both theoretical and empirical fields (Ben-Porat et al., 2019; Hron et al., 2022; Jagadeesan et al., 2023; Meta, 2022; Savy, 2019).

6. **Separation of creators into groups for multi-group setting:** There are $L$ groups of creators, each $\mathcal{C}_\ell \subseteq [n]$, $|\mathcal{C}_\ell| = n_\ell \ge 1$ for $\ell \in [L]$. $\cup_{\ell=1}^L \mathcal{C}_\ell = [n], \cap_{\ell=1}^L \mathcal{C}_\ell = \varnothing$. Group utility for group $\ell$ is denoted as

$$Q_\ell(\boldsymbol{s}) = \sum_{i \in \mathcal{C}_\ell} u_i(\boldsymbol{s}), \text{ where } \boldsymbol{s}_\ell = (\boldsymbol{s}_i)_{i \in \mathcal{C}_\ell}. \quad (4)$$

## 4. Bandit $C^3$ Games

An intriguing property for $C^3$ games is that enhancing one creator's relevance score for a user can potentially decrease

---

[1]Another interpretation of this probability is that the user selects one item from the provided $K$ items according to the Random Utility Model (Yao et al., 2023), where $\beta$ depends on the noise level.

the utilities of other creators about that user. Conversely, decreasing a creator's relevance score may increase the utilities of other creators about that user. This dynamic underscores a fundamental limitation of self-interested utility maximization, which does not necessarily result in optimal group utility. Increases in the relevance score of one creator can detrimentally affect the utilities of others within the group.

Consider a simple scenario, where there are three creators labeled as 1, 2, and 3. Each of them produces videos about basketball under a stable equilibrium—meaning none of them has an individual incentive to switch topics. However, if creator 3 were to change her focus to another topic, e.g., music, this shift could potentially increase the rewards received by creators 1 and 2. Importantly, the additional rewards gained by creators 1 and 2 might surpass the loss in rewards experienced by creator 3. Consequently, although creator 3's utility decreases, the overall utility of the group increases. This illustrates how individual actions within a collaborative environment can impact collective outcomes, sometimes positively, even if they are detrimental to the individual initiating the change.

Building on this intuition, we analyze content distribution, equilibria, and user welfare in a representative class of $C^3$ games, namely the Bandit $C^3$ Game. The Bandit $C^3$ Game models a scenario where the user population comprises multiple interest groups, each with orthogonal preference representations. Each content creator has the option to cater to one and only one user group. While this game is simple and stylized, it simulates the fundamental scenario where every creator must select a topic to create content.

**Definition 4.1** (Bandit $C^3$ Game). The Bandit $C^3$ game is specified by the following RS environments:

- The user population is a distribution on $\{e_j\}_{j=1}^m$ where $x = e_j$ with probability $\{p_j\}_{j=1}^m$ and $E = \{e_1, \cdots, e_m\} \subset \mathbb{R}^m$ is the set of unit basis vectors in $\mathbb{R}^m$.

- Without additional specification, all creators share the same action set $\mathcal{S}_i = E$. The relevance is measured by the inner product, i.e., $\sigma(s; x) = s^\top x$.

We first analyze the scenario in which a single group of creators coexists with individual creators.

### 4.1. Single-group

The pure Nash equilibrium (PNE) (Nash Jr, 1950) is the most widely recognized concept for characterizing the outcome of a game. At a PNE, no player can increase their utility by unilaterally deviating from their current strategy, given the strategies of the other players.

**Definition 4.2** (Vanilla equilibrium). We say $s^v$ is a vanilla

equilibrium if for every $i \in [n], s'_i \in \mathcal{S}_i$,

$$u_i(s^v) \geq u_i(s'_i, s^v_{-i}). \tag{5}$$

In other words, a vanilla equilibrium is an individual PNE in bandit $C^3$ games.

In bandit $C^3$ games, a vanilla equilibrium can be represented as $\{q_j^v\}_{j=1}^m$, where $q_j^{eq}$ denotes the number of creators selecting $e_j$ in a specific equilibrium $eq$. With a little abuse of notation, we use $s = s'$ to denote the profile of two equilibria $s$ and $s'$ shapes the same, i.e., they are equal up to a permutation of strategies.

When there is only one group, i.e., $|\mathcal{C}_1| = n_c$ and $|\mathcal{C}_2| = |\mathcal{C}_3| = \cdots = |\mathcal{C}_L| = 1$, we denote this group of creators $\mathcal{C}_1$ as a set $\mathcal{C} \subseteq [n]$, where $|\mathcal{C}| = n_c$. The group utility for $\mathcal{C}$ is denoted as $Q_c(s)$. For a fixed group strategy $s_c$ in a $C^3$ game $\mathcal{G}$, we denote the set of PNE for the remaining players as $\mathrm{PNE}(\mathcal{G}, s_c)$. We can prove that a PNE and a $\mathrm{PNE}(\mathcal{G}, s_c)$ always exist in bandit $C^3$ games.

**Lemma 4.3.** *Any bandit $C^3$ game has a vanilla equilibrium and a $\mathrm{PNE}(\mathcal{G}, s_c)$ for any fixed group strategy $s_c$.*

Ideally, this group of creators can optimize their group utility while considering the dynamics of other individual creators. Based on Lemma 4.3, we know that a Nash equilibrium for individual creators outside the group always exists when the group strategy is fixed. This allows us to define a Stackelberg-type equilibrium for this game. Its formal definition is as follows.

**Definition 4.4** (Single group Stackelberg equilibrium). We denote $s_c^{\mathrm{I}}$ as the solution for the following bilevel optimization:

$$\max_{s_c} Q_c(s_c, s_{-c}), \quad \text{s.t.} \quad s_{-c} \in \mathrm{PNE}(\mathcal{G}, s_c).$$

And $s_{-c}^{\mathrm{I}} = \mathrm{PNE}(\mathcal{G}, s_c^{\mathrm{I}})$. Here, $s_{-c}$ is selected from $\mathrm{PNE}(\mathcal{G}, s_c)$ according to a specific tie-breaking rule, such as a creator choosing the user with the smaller index when the utilities for selecting different users are equal. We name $s^{\mathrm{I}}$ as Type-I equilibrium, which denotes the single group Stackelberg equilibrium.

We also consider the single-group PNE of this game by treating the group as a single entity. A single-group PNE is a strategic profile where no entity, including both the group and individual creators, can improve its utility via unilateral deviation. A single-group PNE does not necessarily exist in a bandit $C^3$ game.

**Definition 4.5** (Single-group Nash equilibrium). Suppose $s^{\mathrm{II}}$ is a Single-group Nash Equilibrium. Then, for any $s'_c$ and any $i \notin \mathcal{C}$, $s^{\mathrm{II}}$ satisfies the following conditions:

$$Q_c(s^{\mathrm{II}}) \geq Q_c(s'_c, s^{\mathrm{II}}_{-c}), \text{ and } u_i(s^{\mathrm{II}}) \geq u_i(s'_i, s^{\mathrm{II}}_{-i}).$$

We refer to the joint strategy profile $s^{\mathrm{II}}$ as a Type II.

## 4.2. Type $\mathrm{I}, \mathrm{II} =$ vanilla equilibrium

*Example* 1. $m = 2$, $n = 5$, $p_1 = 0.7, p_2 = 0.3$, $\mathcal{C} = \{1, 2\}$. $s^{\mathrm{v}} = s^{\mathrm{I}} = \{s_{\{1,2,3\}} = e_1, s_{\{4,5\}} = e_2\}$. $s^{\mathrm{II}} = \{s_{\{1,2,3\}} = e_1, s_{\{4,5\}} = e_2\}$.

*Example* 2. $m = 3$, $n = 3$, $p_1 = 0.5, p_2 = 0.4, p_3 = 0.1$. $\mathcal{C} = \{1, 2\}$. $s^{\mathrm{v}} = \{s_{\{1,2\}} = e_1, s_3 = e_2\}$. $s^{\mathrm{I}} = \{s_1 = e_1, s_2 = e_3, s_3 = e_2\}$. $s^{\mathrm{II}} = \{s_1 = e_1, s_2 = e_2, s_3 = e_1\}$.

We use two example games to illustrate whether Type I is a vanilla equilibrium. When considering the dynamic strategic moves of out-of-group creators, some improvements to group strategies may ultimately fail to increase the group's utility. In Example 1, we first consider equilibrium I, where the group has the incentive to change the strategy. If they adjust $s_2 = e_1 \to s'_3 = e_2$, then creator 4 has the incentive to change its strategy from $s_4 = e_2 \to s'_4 = e_1$, and this change will decrease the group utility. After these updates, the group utility decreases. Therefore, the original strategy profile remains as equilibrium I. This is because individual creators who select $e_2$, facing a change $s_2 = e_1 \to s'_2 = e_2$, are influenced to shift strategies due to increased rewards for $e_1$ and reduced rewards for $e_2$. Thus, considering the strategic moves of individual creators, the group will not change its strategy. In contrast, in Example 2, under the vanilla equilibrium profile, the group changes their strategy from $s_2 = e_1 \to s'_2 = e_3$. After this change, the individual creator 2 has no incentive to change its strategy, and the group utility increases, indicating that equilibrium I differs from the vanilla equilibrium. Based on these observations, we develop constraints to ensure that Type I and Type II align with the vanilla equilibrium.

**Theorem 4.6.** *Under a specific tie-breaking rule, if a bandit $C^3$ game admits a vanilla equilibrium $s^{\mathrm{v}}$, a Type I equilibrium $s^{\mathrm{I}}$, a Type II equilibrium $s^{\mathrm{II}}$, and $n_c \leq \min_{j \in [m]} q_j^{\mathrm{v}}$, then $s^{\mathrm{I}} = s^{\mathrm{II}} = s^{\mathrm{v}}$.*

A specific tie-breaking rule can be a creator will choose the user with the smaller index when the utilities for selecting different users are equal. This theorem demonstrates that, when the group size is small, group behavior does not affect the equilibrium of the bandit $C^3$ game. This is primarily because, when a group creator attempts to strategically change its topic to enhance group utility, other individual creators will fill the resulting gap, as demonstrated in Example 1.

When the group size is small and there are no niche topics (a small $p_j$), group behavior does not influence the equilibrium profile. However, in many practical cases, such as in TvN games in the following analysis, the group strategies have a significant influence on content distribution and user welfare.

## 4.3. Type $\mathrm{I}, \mathrm{II} \neq$ vanilla equilibrium

**TvN (Trend v.s. Niche) (Yao et al., 2024a)** TvN game is first introduced by Yao et al. (2024a), where $p_1 = \frac{n+1}{2n}$, and $p_2 = \cdots = p_n = \frac{1}{2n}$. The TvN game captures the essence of real-world user populations and the dilemmas faced by creators. Creators often find themselves at a crossroads: they must decide whether to pursue popular trends to reach a broader audience, which leads to intense competition, or focus on niche topics with a smaller audience and reduced competition.

Next, we use this TvN game to illustrate the difference in user welfare between the group case and the individual case.

**Theorem 4.7.** *In a TvN game with exposure rewards, we have $s^{\mathrm{I}} = s^{\mathrm{II}}$. If $r_1 = r_2 = \cdots = r_K = 1$, the following results hold:*

1. *Individual case ($n_c = 1$):*

$$W(s^{\mathrm{v}}, K) = \frac{n+1}{2n} K,$$

*and the optimal welfare is*

$$W_{\max} = \max_{s, K} W(s, K) = \frac{n+1}{2}.$$

2. *Full group case ($n_c = n$):*

$$W(s^{\mathrm{II}}, K) = 1.$$

3. *Partial group case ($1 < n_c < n$): If $\beta \to 0$, then*

$$q_1^{\mathrm{II}} \leq \max \left\{ 1, \left\lceil \sqrt{(n+1)(n-n_c)} \right\rceil \right\},$$

*and the welfare loss is*

$$W_{\max} - W(s^{\mathrm{II}}) = \frac{n - q_1^{\mathrm{II}}}{2}.$$

*If $\beta \to \infty$, then $q_1^{\mathrm{II}} = n$, and $W_{\max} = W(s^{\mathrm{II}})$.*

We present the results for the case where users have *constant attention* over the recommended content, i.e., $r_1 = r_2 = \cdots = r_K = 1$. This attention is truncated by the parameter $K$ and it assumes users allocate a fixed amount of attention uniformly across the top-$K$ items they are shown. This setting is relevant in practice, particularly in user interfaces where content is displayed in fixed-sized, unordered blocks (e.g., a "For You" page with $K$ equally weighted items). These findings can be readily extended to the general case of $\{r_i\}_{i=1}^n$ through straightforward adaptation.

To emphasize the potential negative impact of group strategic behavior on user welfare, we consider the TvN game as an example for worst-case analysis (Yao et al., 2024a).

The user proportion $p$ in the TvN game, although stylized, reflects user distributions in real-world online content platforms, which are often highly skewed and unbalanced. And our choice of this $p$ is also primarily for clarity and simplification of presentation. We extend our results to a more general unbalanced case in the Appendix A.4, where the largest user proportion in the TvN game can vary, and this will provide a more smooth transit from the extreme case to the even case.

In this setting, the formation of groups can significantly impact user welfare. The content creation game with exposure rewards exhibits a nearly zero-sum structure. When all creators form a single group, they have no incentive to compete internally, and the optimal group strategy becomes $\{s_i = \mathbf{e}_i\}_{i=1}^n$. This strategy leads to a drastic decline in user welfare compared to the optimal welfare scenario. This result indicates that the worst-case scenario for user welfare occurs when a group of creators dominates the entire exposure to certain users or topics, which means these users are only exposed to content created by this group.

The scenario where all creators form a group has practical relevance in real-world cases. For instance, in online media platforms, certain modules restrict user access to content creators within a specific geographic proximity (e.g., the "People Nearby" feature). In such cases, it is reasonable to assume that all creators within a small region may form a group, especially when the creator population is limited. Furthermore, joining a group is a dominant strategy, as the group utility is non-decreasing compared to the individual case—at the very least, it remains unchanged if creators maintain their previous actions.

*Remark* 4.8. In a TvN game with exposure rewards, adjustments to the parameters $K$ and $\beta$ cannot influence individual creators' decisions to select $\mathbf{e}_1$ and chase the trend (Yao et al., 2024a). However, when considering creators' group behavior, the platform can adjust $K$ and $\beta$ to improve user welfare. Increasing $\beta$ results in a more uniform distribution of user $\mathbf{x}$'s matches across the top-$K$ candidates, while increasing $K$ expands the set of creators that user $\mathbf{x}$ is exposed to (Yao et al., 2024c), so both of them introduce additional competition into the game. As shown in Theorem 4.7 item 3, under constant attention scores, additional competition leads to improved user welfare outcomes. For general $\{r_i\}_{i=1}^n$, the user welfare is given by

$$W(\mathbf{s}) = \frac{n+1}{2n} \sum_{i=1}^{\min(K, q_1)} r_i + \frac{n - q_1}{2n} r_1.$$

By tuning $K$ and $\beta$, the platform can regulate the number of creators within the group who select $\mathbf{e}_1$, thereby enhancing user welfare. More generally, the platform can strategically set values for $K$ and $\beta$ to achieve a limited improvement on user welfare by steering the equilibrium of the game.

**Theorem 4.9.** *In a TvN game with exposure reward, $r_1 = r_2 = \cdots = r_\tau = 1$, $r_{\tau+1} = \cdots = r_n = 0$, $\beta$ is sufficiently large, $\mathcal{S}_i = E \setminus \{\mathbf{e}_1\}$ for $i \notin \mathcal{C}$, $\mathcal{S}_i = E$ for $i \in \mathcal{C}$, and the group size $n_c > \tau$, then*

$$W(\mathbf{s}^{\mathrm{II}}, K) = \begin{cases} \frac{n+1}{2n} K + \frac{n-K}{2n}, & K \leq \tau, \\ \frac{n+1}{2n} \tau + \frac{n-K}{2n}, & \tau + 1 \leq K \leq n_c. \end{cases}$$

Theorem 4.9 considers another setting with *diminishing attention*, modeled as $r_1 = \cdots = r_\tau = 1$, $r_{\tau+1} = \cdots = r_n = 0$, which captures the case where users only pay attention to the top $\tau$ items. These two types—constant (Theorem 4.7) and diminishing—represent common user behaviors corresponding to slow-decay and rapid drop-off attention curves, respectively.

In this game, individual creators are restricted from selecting $\mathbf{e}_1$, reflecting real-world scenarios where such limitations arise. Potential reasons for this include: (1) They lack expertise in $\mathbf{e}_1$, making the cost of creating content related to $\mathbf{e}_1$ prohibitively high. (2) Due to their local updates, they only explore a limited region of the strategy space. The group will assign at most $K$ creators to $\mathbf{e}_1$ to capture all the exposure associated with $\mathbf{e}_1$. This result also provides insight into the scenario where a single group specializes in a specific topic and monopolizes the associated exposure. In such cases, the group can reduce its collective effort on the topic while still capturing the entirety of the exposure.

For the first part of Theorem 4.9, when $K \leq \tau$, increasing $K$ under constant attention scores leads to improved user welfare. For the second part of Theorem 4.9, when $\tau + 1 \leq K \leq n_c$, the results reveal a counterintuitive phenomenon: in a top-$K$ recommendation system, user welfare does not necessarily increase with larger values of $K$. This contrasts with established findings that user welfare typically improves as $K$ increases (Yao et al., 2023) and provides insight that platforms should avoid selecting excessively large $K$ in recommendation systems where users have diminishing attention spans. Additionally, we provide a game instance under the low attention spans of users, demonstrating that a higher $\beta$ does not always lead to better user welfare. The proof follows a similar approach to the proof of Theorem 4.7, item 3.

**Corollary 4.10.** *In a TvN game with exposure rewards, if $r_1 = 1$ and $r_2 = \cdots = r_n = 0$, and when $1 < n_c < n$,*

$$W(\mathbf{s}^{\mathrm{II}}, \beta=0) - W(\mathbf{s}^{\mathrm{II}}, \beta=\infty) = \frac{n - q_{1,0}^{\mathrm{II}}}{2n}.$$

*where $q_{1,0}^{\mathrm{II}} \leq \max \left\{ 1, \lceil \sqrt{(n+1)(n-n_c)} \rceil \right\}$.*

### 4.4. Multi-groups in bandit $C^3$ games

For bandit $C^3$ games with multiple groups, the results are similar to those in the single-group case. We briefly present

these results here, as they share similar intuition and insights.

**Definition 4.11** (Multi-group PNE). For any $\ell \in [L]$, $s^{\mathrm{g}}$ is a multi-group PNE if for any $s'_\ell$,

$$Q_\ell(s'_\ell, s^{\mathrm{g}}_{-\ell}) \leq Q_\ell(s^{\mathrm{g}}).$$

**Theorem 4.12.** *In a bandit $C^3$ game with $\beta \to 0$, suppose a vanilla equilibrium is $\{q^{\mathrm{v}}_j\}^m_{j=1}$. If $\max_{\ell \in [L]} n_\ell \leq \min_{j \in [m]} q^{\mathrm{v}}_j$, then for any multi-group PNE $s^{\mathrm{g}}$ in this game, we have for any $j \in [m]$,*

$$q^{\mathrm{g}}_j - q^{\mathrm{v}}_j \leq 2, q^{\mathrm{v}}_j - q^{\mathrm{g}}_j \leq \min\left\{1 + \max_{r \neq s} \frac{p_r}{p_s}, \frac{1}{3} q^{\mathrm{v}}_j\right\}.$$

The proof is in Appendix A.7. Similar to Theorem 4.6, Theorem 4.12 demonstrates that the multi-group PNE (if it exists) is unlikely to differ significantly from the vanilla equilibrium when the group size is small. We also introduce a Restricted Multi-Trend vs. Niche (MTvN) game in Appendix A.6, a variant of the TvN game adapted to the multi-group setting. This game exhibits results similar to those in the single-group TvN game.

## 5. General Case and PoA

In this section, we analyze user welfare under any possible creator group partition and group coarse correlated equilibrium (GCCE) in the $C^3$ game through the lens of the Price of Anarchy (PoA) (Koutsoupias & Papadimitriou, 1999). The PoA quantifies the inefficiency in user welfare resulting from creators' strategic group behaviors and self-interested actions.

**Definition 5.1** (GCCE). $\alpha$ is a group CCE, for every group $\ell \in [L]$ and every group strategy $s'_\ell$,

$$\mathbb{E}_{s \sim \alpha}[Q_i(s)] \geq \mathbb{E}_{s \sim \alpha}[Q_i(s'_\ell, s_{-\ell})]. \tag{6}$$

**Definition 5.2** (PoA under GCCE). Define the price of anarchy of a $C^3$ instance $\mathcal{G}$ as

$$PoA(\mathcal{G}) = \frac{\max_s W(s)}{\min_{\alpha \in \mathrm{GCCE}(\mathcal{G})} \mathbb{E}_{s \sim \alpha}[W(s)]}, \tag{7}$$

where GCCE($\mathcal{G}$) is the set of GCCEs of $\mathcal{G}$ for all possible creator group partitions.

Based on the definition, $PoA(\mathcal{G}) \geq 1$, and larger values indicate worse user welfare. The definition of PoA under GCCE (Blum et al., 2008) provides the strongest possible welfare guarantee. Specifically, any upper bound on the Price of Anarchy (PoA) under GCCE automatically applies to the PoA under more refined equilibrium concepts, such as group CE, group PNE, or mixed group NE (if they exist), as these equilibria are all special cases of GCCE.

As the previous section indicates, when the group is big, especially when all creators form as a group or multiple groups dominate multiple topics respectively, we show that group strategies can inevitably lead to a big fraction loss of the optimal welfare. Now we show that in the worst case, PoA in a $C^3$ game with exposure reward can be arbitrarily bad.

**Theorem 5.3.** *For any $C^3$ instance $\mathcal{G}$ with exposure rewards,*

$$PoA(\mathcal{G}) \to \infty.$$

This phenomenon occurs when a group dominates the total exposure to a user, even though they reduce their effort and produce low-quality content ($s_i = \varepsilon \mathbf{1}$, $\varepsilon$ is an arbitrarily small value), they still dominate the exposure to this user. Both Theorem 4.7 and 5.3 indicate that the most severe issue with exposure-based rewards occurs when a group of creators monopolizes the total exposure to users or when only a group of creators specializes in specific topics. As highlighted in Yao et al. (2023), rewarding creators based on user engagement is likely to improve the user welfare of recommendations. In this work, the engagement reward for creator $i$ is defined as follows:

Engagement reward: $u_i(s) = \mathbb{E}_{x \in \mathcal{X}}\left[\pi(x, s)P_i(s, x)\right]$.

Notably, when user engagement is used as the reward mechanism, the welfare of creators aligns with user welfare. Consequently, when all creators act as a group, the optimal group strategy also maximizes user welfare, which means PoA $= 1$ in this case. This engagement reward mitigates the user welfare loss caused by large or professional groups, leading to optimal user welfare outcomes in the TvN and MTvN games discussed in previous sections. However, we provide a negative result on the PoA bound under the user engagement reward mechanism with constant user attention and construct a lower-bound instance to demonstrate the tightness of this bound.

**Theorem 5.4** (PoA under engagement reward). *For any $C^3$ instance $\mathcal{G}$ with engagement rewards, $\beta \geq 1$, $1 \leq K \leq n$, and user welfare defined as in Eq. (2), if $r_1 = \cdots = r_K = 1$, then*

$$PoA(\mathcal{G}) \leq 1 + K.$$

The proof of Theorem 5.4 is provided in Appendix B.2. One important observation regarding the bound is that the benchmark $\max_s W(s, K)$ varies with $K$. As $K$ increases, the benchmark $\max_s W(s, K)$ may also increase. Although the PoA of the $C^3$ game is no longer infinite, it still suffers from a significant loss in user welfare. We next conduct a worst-case analysis to identify the scenarios that lead to this unsatisfactory PoA result.

**Theorem 5.5** (Lower Bound). *Given any* $1 \leq K \leq n$, *$\beta \geq 1$, and $r_1 = \cdots = r_K = 1$, there exists a $C^3$ game instance with engagement rewards such that*

$$PoA(\mathcal{G}) \geq 1 + (K-1)e^{-\frac{1}{\beta}}.$$

The proof of Theorem 5.5 is provided in Appendix B.3, which explicitly constructs a game instance that achieves the stated PoA lower bound. In this instance, the vanilla equilibrium occurs when all creators in the game select a user with a very small proportion. This leads to a high utility for that user, which in turn results in high creator utility. Consequently, no creator has an incentive to deviate from their strategy. This leads to a suboptimal outcome for user welfare, but this worst-case scenario can be alleviated by the approach proposed in Yao et al. (2024c), which helps creators escape from the current suboptimal state through a user importance reweighting method.

These results collectively provide insights for online content platforms to mitigate user welfare loss caused by group strategies: use user engagement-based rewarding mechanisms and implement intervention methods such as the user importance reweighting method proposed in Yao et al. (2024c). The importance reweighting method enables the platform to steer creator incentives toward under-served users by modifying the reward structure. Specifically, the platform defines the creator's utility as

$$u_i(\boldsymbol{s}) = \mathbb{E}_{\boldsymbol{x} \in \mathcal{X}}\left[w(\boldsymbol{x})\pi(\boldsymbol{x}, \boldsymbol{s})P_i(\boldsymbol{s}, \boldsymbol{x})\right],$$

where $w(\boldsymbol{x})$ represents the *importance weight* of user $\boldsymbol{x}$. When the platform detects that a user is being under-served under the current content distribution, it increases $w(\boldsymbol{x})$ for that user. This effectively amplifies the reward for creators who target such users, encouraging them to shift their content in that direction. Over time, this reshapes the content distribution and improves overall user welfare.

We also show that in some cases, such as when $\beta \to 0$ and relevance scores and user attention scores are binary (i.e., $\sigma \in \{0, 1\}$, $r_1 = r_2 = \cdots = r_\tau = 1$, and $r_{\tau+1} = \cdots = r_n = 0$), we obtain a positive result on the PoA.

**Proposition 5.6** (PoA under binary $\sigma$). *For any $C^3$ instance $\mathcal{G}$ with engagement rewards, $\beta \to 0$, and user welfare defined as in Eq. (2), if the relevance scores and user attention scores are binary (i.e., $\sigma, r_i \in \{0, 1\}$), then*

$$PoA(\mathcal{G}) \leq 2.$$

## 6. Simulations

**Synthetic Environment**   For the synthetic environment, we first construct the user population as follows: we fix an embedding dimension $d = 5$ and independently sample $m = 10$ users from the unit sphere $\mathbb{S}^{d-1}$. The distribution of the 10 users is $\boldsymbol{p} = \frac{1}{200} \times [100, 50, 20, 10, 10, 5, 2, 1, 1, 1]^\top$. We then slightly polarize the user distribution to simulate a real-world scenario (Dean & Morgenstern, 2022; Lin et al., 2024), where the largest user group is positioned at one pole and has some distance from the other users. This setup presents a more challenging case for user welfare optimization, as in some scenarios, creators must traverse a long path from one pole to the other to improve user welfare.

The relevance score function is set to $\sigma(\boldsymbol{s}, \boldsymbol{x}) = \frac{1}{2}(\boldsymbol{s}^\top \boldsymbol{x} + 1) \in [0, 1]$. We use 2 kinds of user attention scores: constant attention scores where $r_i = 1$ for $i \in [5]$ and log cutoff attention scores $\{r_i\}_{i=1}^n = \{\frac{1}{\log_2 2}, \frac{1}{\log_2 3}, \cdots, \frac{1}{\log_2 6}, 0, \cdots, 0\}$. We set $(\beta, K) = (0.1, 5)$ by default. This synthetic dataset characterizes a class of clustered user preference distributions, such as majority versus minority user groups.

On the creators' side, there are $n = 30$ creators in total, including one group of creators. We vary the group size $n_c \in \{10, 15, 20, 25, 30\}$. The group of creators' strategies is initialized near user $\boldsymbol{x}_1$, while the remaining creators' strategies are initialized near the other users. This setup models a scenario where the group of creators specializes in a specific topic. Individual creators use the LBR algorithm (Yao et al., 2024c;a) to update their strategies, while the group performs multiple steps of gradient descent on the objective $Q_c(s_c, s_{-c})$ with respect to the group strategy $s_c = (s_1, s_2, \cdots, s_{n_c})$.

**Results**   We analyze the average user welfare under varying reward types, group sizes, and attention score distributions, with a fixed time horizon of $T = 100$. The results are presented in Figure 1. Our experiments reveal that user welfare under engagement-based rewards consistently surpasses that under exposure-based rewards. Notably, user welfare experiences a significant decline when the group size increases to $n_c = 25$ or $30$ under exposure rewards, whereas it remains stably high under engagement rewards. This observation is further supported by a visualization example in Appendix C.2 ($n_c = 10$), which illustrates that creators in the exposure reward setting do not overlap with the largest user segment $\boldsymbol{x}_1$, as they already dominate the exposure of $\boldsymbol{x}_1$. In contrast, creators under engagement rewards exhibit overlap with $\boldsymbol{x}_1$, leading to enhanced user welfare compared to the exposure reward scenario. Our experiments also demonstrate that the PoA for engagement rewards remains close to 1. This result is supported by the theoretical bounds $\max_{\boldsymbol{s}} W(\boldsymbol{s}) \leq 5$ in the constant attention case and $\max_{\boldsymbol{s}} W(\boldsymbol{s}) \leq \sum_{i=1}^5 \frac{1}{\log_2 i+1} < 2.95$ in the general case, highlighting the effectiveness of engagement rewards.

We conduct additional simulations with varying values of $K$ and $\beta$ to validate two key theoretical results presented in Section 4. The outcomes are reported in Appendix C.3 and

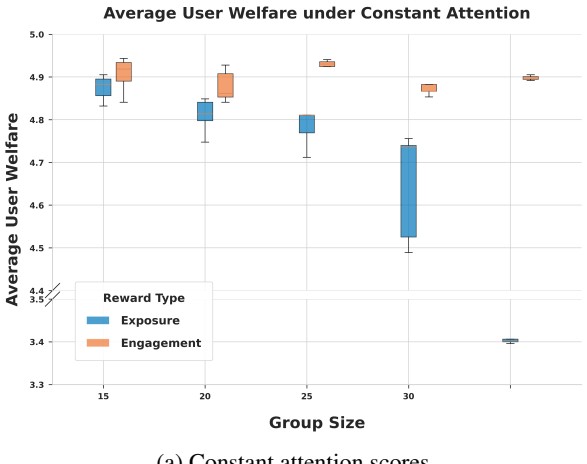

(a) Constant attention scores

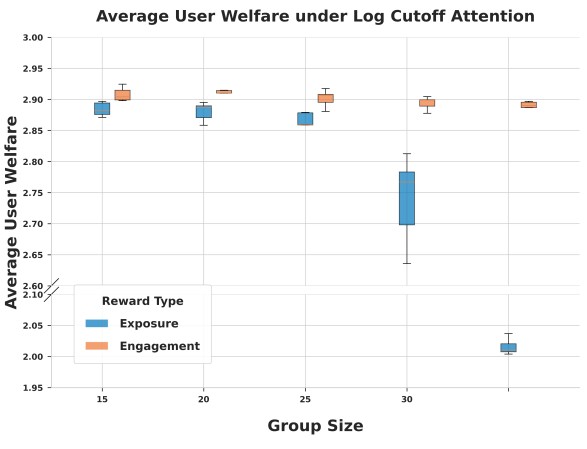

(b) Log cutoff attention scores

Figure 1: User welfare under different reward types, group sizes, and attention scores.

Appendix C.4. We further evaluate the robustness of our findings under a more general user distribution $p$ following a Zipf-like pattern, as detailed in Appendix C.5. Additionally, we consider an alternative creator initialization, where all creators are initialized around users 2 to 10. The results of this setting are provided in Appendix C.6.

## 7. Conclusion

Our work sheds light on the critical implications of group strategies among content creators in online recommendation platforms. Using a game-theoretic framework, we demonstrate how group strategies reshape content distribution and user welfare. While small groups exhibit limited effects, large groups can significantly alter content distribution, leading to substantial user welfare losses. We analyze the role of key parameters $K$ and $\beta$ in exposure rewards and investigate a user engagement-based rewarding mechanism, demonstrating their potential to mitigate the adverse effects of strategic group behavior. These findings offer valuable insights for designing recommendation systems that enhance user welfare while addressing challenges posed by strategic group behavior.

## Acknowledgments

The research work described in this paper was conducted in the JC STEM Lab of Machine Learning and Symbolic Reasoning funded by The Hong Kong Jockey Club Charities Trust. The authors would like to thank Hanrui Zhang for valuable discussions and feedback.

## Impact Statement

This paper presents work whose goal is to advance the field of game theory for recommendation systems. There are many potential societal consequences of our work, none of which we feel must be specifically highlighted here.

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

# Appendix

## A. Proofs for Section 4

### A.1. Proof for Lemma 4.3

*Proof.* We prove the existence of a single-group pure Nash equilibrium (PNE) by constructing it using a self-greedy algorithm. Suppose creators select strategies sequentially, and let:

- $q_i^t$ denote the number of creators selecting $e_i$ at time $t$. The final number of creators selecting $e_i$ is $q_i^T$, where $T = n$.

- $v_i(q_i)$ represent the utility for selecting $e_i$, defined as:

$$v_i(q_i) = \begin{cases} \frac{p_i \exp(\beta^{-1})}{q \exp(\beta^{-1}) + (K-q)}, & 1 \le q_i \le K, \\ \frac{p_i}{q_i}, & q_i \ge K+1. \end{cases}$$

  Note that $v_i(q_i)$ is non-increasing in $q_i$. When more than $K$ creators have identical relevance scores for user $i$, ties are broken by uniformly and randomly selecting $K$ creators, ensuring $v_i(q_i) = \frac{p_i}{q_i}$ for $q_i \ge K+1$.

- The equilibrium condition requires:

$$v_i(q_i^T) \ge v_j(q_j^T + 1) \quad \text{for all } i \ne j.$$

We construct the equilibrium using a self-greedy algorithm, where creators sequentially select strategies to maximize their utility. The algorithm proceeds as follows:

---
**Algorithm 1** Self-Greedy Algorithm for Constructing a PNE

---
**Require:** Number of creators $n$, initial counts $q_i^0 = 0$ for all $i$.
 1: **for** $t = 1$ **to** $n$ **do**
 2:     $s_t = \text{argmax}_{i \in [m]} v_i(q_i^{t-1} + 1)$
 3: **end for**

---

Let $t_i^0$ denote the time when the last creator selects $e_i$, i.e., $t_i^0 = \text{argmin}_t q_i^t = q_i^T$. The algorithm ensures:

$$v_i(q_i^T) = v_i(q_i^{t_i^0}) \ge v_j(q_j^{t_i^0} + 1) \quad \text{for all } j \ne i.$$

Since $v_j(q_j)$ is non-increasing in $q_j$, we have $v_j(q_j^{t_i^0} + 1) \ge v_j(q_j^T + 1)$ because $q_j^{t_i^0} \le q_j^T$. Thus, $\{q_i^T\}_{i=1}^m$ constitutes a pure Nash equilibrium.

To extend this result to a group of creators with fixed strategies $s_c$, we revise the initial conditions $\{q_i^0\}_{i=1}^m$ and repeat the procedure. This yields a PNE for the out-of-group creators, denoted as $\text{PNE}(s_c)$. □

### A.2. Proof for Theorem 4.6

*Proof.* We assume a Type I is $\{q_j\}_{j=1}^m$. If $\{q_j\}_{j=1}^m \ne \{q_j^v\}_{j=1}^m$, there must exists $j \in [m]$ $q_j > q_j^v$, then there exists $k \ne j$, $q_k < q_k^v$, and we have

$$v_k(q_k + 1) \ge v_k(q_k^v) \ge v_j(q_j^v + 1) \ge v_j(q_j).$$

Since $n_c \le \min_{j \in [n]} q_j$, then there exists $i \in [n]/\mathcal{C}$, $s_i = e_j$. If $v_k(q_k^v) > v_j(q_j^v + 1)$, then creator $i$ will have incentive to change its action from $e_j$ to $e_k$, so $\{q_j\}_{j=1}^m$ is not an Type I. If $v_k(q_k^v) = v_j(q_j^v + 1)$, by the assumption that $\{q_j^v\}_{j=1}^m$ is formed by some specific tie-breaking rules, meaning that creators prefer selecting $e_k$ to $e_j$, then creator $i$ will change its action from $e_j$ to $e_k$, so $\{q_j\}_{j=1}^m$ is not an Type I. By contradiction, Type I is a vanilla equilibrium. Analogously, Type II is also a vanilla equilibrium.

□

**Lemma A.1.** *In the TvN game, for any equilibrium $s^{\mathrm{v}}$, $s^{\mathrm{I}}$, or $s^{\mathrm{II}}$, the following conditions on the $q_i$ values hold:*

$$q_1 \geq 1, \quad q_i \in \{0, 1\} \text{ for } i \neq 1.$$

*Additionally, the strategy of any individual creator is $e_1$, and we have $s^{\mathrm{I}} = s^{\mathrm{II}}$.*

*Proof.* In the TvN game, the probabilities of exposure are given by:

$$p_1 = \frac{n+1}{2n}, \quad p_i = \frac{1}{2n} \text{ for } i \neq 1.$$

The reward for selecting strategy $e_i$ is determined as follows:

$$v_i(q_i) = \begin{cases} 0, & \text{if } q_i = 0, \\ \frac{p_i \exp(\beta^{-1})}{q \exp(\beta^{-1}) + (K-q)}, & \text{if } 1 \leq q_i \leq K, \\ \frac{p_i}{q_i}, & \text{if } q_i \geq K+1. \end{cases}$$

.

Now, suppose $q_1 = 0$. In this case, there must exist distinct indices $i \neq j$ such that $s_i = s_j = e_k$, with corresponding utilities $u_i$ and $u_j$. If $s_i$ changes its action from $e_k$ to $e_1$, the new utilities of the agents are denoted as $u_i'$ and $u_j'$, respectively. From this, we obtain the following inequalities:

$$u_i' > u_i, \quad u_j' \geq u_j, \quad u_i' + u_j' > u_i + u_j.$$

Thus, in $s^{\mathrm{v}}$, $s^{\mathrm{I}}$, or $s^{\mathrm{II}}$, we conclude that $q_1 \geq 1$.

Next, suppose $s_i = s_j = e_k$, with utilities $u_i$ and $u_j$ for agents $i$ and $j$, respectively. Since $q_1 \geq 1$, there must exist a $k' \neq 1, k$ such that $q_{k'} = 0$. If $s_i$ changes its action from $e_k$ to $e_{k'}$, the new utilities of the agents are denoted by $u_i'$ and $u_j'$, respectively. Again, the following inequalities hold:

$$u_i' > u_i, \quad u_j' \geq u_j, \quad u_i' + u_j' > u_i + u_j.$$

Thus, in $s^{\mathrm{v}}$, $s^{\mathrm{I}}$, or $s^{\mathrm{II}}$, it follows that $q_i \leq 1$ for all $i \neq 1$.

Finally, in the TvN game, for any $K \leq n$ and $i \neq 1$, we have:

$$v_1(q_1) \geq v_1(n) = \frac{n+1}{2n^2} > \frac{1}{2n} \geq v_i(1).$$

Therefore, any individual creator will choose $e_1$ as their strategy. Since individual creators will all select $e_1$, it follows that $s^{\mathrm{I}} = s^{\mathrm{II}}$. $\qquad\square$

## A.3. Proof for Theorem 4.7

*Proof.* (1) When $n_c = 1$, Based on Lemma A.1, $\quad q_i \in \{0, 1\}$ for $i \neq 1$, so the general user welfare is

$$W(s) = \frac{n+1}{2n} \sum_{i=1}^{\min(K, q_1)} r_i + \frac{n - q_1}{2n} r_1. \tag{8}$$

According to Lemma A.1, for the vanilla equilibrium $s^{\mathrm{v}} = \{s_i = e_1 \text{ for all } i \in [n]\}$, for any $\{r_i\}_{i=1}^n$, the optimal welfare is achieved when $K = n$:

$$\operatorname*{argmax}_K W(s^{\mathrm{v}}, K) = n.$$

And when $r_1 = r_2 = \cdots = r_n = 1$,

$$W(s) = \frac{n+1}{2n} \min(K, n),$$

So $W(s)$ is maximized when $K = n$, and $W_{\max} = \frac{n+1}{2}$.

(2) When $n_c = n$, we have

$$Q_c(s) \leq 1$$

The equality in the above inequality holds when

$$s^{\mathrm{I}} = s^{\mathrm{II}} = \{s_i = e_i \text{ for } i \in [n]\}.$$

Under this equilibrium, the user welfare $W(s^{\mathrm{II}}) = 1$.

(3) When $\beta \to 0$,

$$v_i(q_i) = \begin{cases} 0, & \text{if } q_i = 0, \\ \frac{p_i}{q_i}, & \text{if } q_i \geq 1. \end{cases}$$

Let the number of creators in the group choosing $e_1$ be $q_1(c)$, $q_1(c) \geq 1$, and we have

$$Q_c(q_1(c)) = \frac{n+1}{2n} \cdot \frac{q_1(c)}{n - n_c + q_1(c)} + \frac{1}{2n} \cdot (n_c - q_1(c)).$$

Maximize $Q_c(q_1(c))$ and we got

$$q_1(c)^\star = 1 \text{ or } \lfloor \sqrt{(n+1)(n - n_c)} - (n - n_c) \rfloor \text{ or } \lceil \sqrt{(n+1)(n - n_c)} - (n - n_c) \rceil.$$

When $\beta \to \infty$,

$$v_i(q_i) = \begin{cases} 0, & \text{if } q_i = 0, \\ \frac{p_i}{n}, & \text{if } q_i \geq 1. \end{cases}$$

$$Q_c(q_1(c)) = \frac{n+1}{2n} \cdot \frac{q_1(c)}{n} + \frac{1}{2n} \cdot \frac{n_c - q_1(c)}{n}.$$

Maximize $Q_c(n_1)$ and we got $q_1(c)^\star = n_c$. Combining $q_1(c)^\star$ with Eq. 8, we complete the proof.

$\square$

### A.4. Additional results for Theorem 4.7 with a more general $p$

**Theorem A.2.** *Consider a bandit $C^3$ game with exposure rewards, where the user distribution satisfies $p_1 \geq \frac{1}{n}$, $p_1 + (n - 1)p_2 = 1$, and $p_2 = p_3 = \cdots = p_n$. Let $\rho_{1,2} = \frac{p_1}{p_2}$. Then, $s^{\mathrm{I}} = s^{\mathrm{II}}$. Moreover, if $r_1 = r_2 = \cdots = r_K = 1$, the following results hold:*

1. ***Individual case** ($n_c = 1$): If $\beta \to 0$, the vanilla equilibrium satisfies $\rho_{1,2} - 1 \leq q_1^v \leq \rho_{1,2}$, and*

$$W(s^v, K) = p_1 q_1^v + p_2(n - q_1^v) \geq (n^2 + 2n - 1)p_2 + \frac{1}{p_2} - 2n - 1.$$

   *If $\beta \to \infty$,*
$$W(s^v, K) = p_1 K,$$

   *and the optimal welfare is $W_{\max} = \max_{s,K} W(s, K) = p_1 n$.*

2. ***Full group case** ($n_c = n$): When all creators form a single group,*

$$W(s^{\mathrm{II}}, K) = 1.$$

3. ***Partial group case** ($1 < n_c < n$): If $\beta \to 0$, two cases arise: (i) If $q_1^v \leq n - n_c$, which is guaranteed when $p_2 \geq \frac{1}{2n - n_c - 1}$, then $q_1^{\mathrm{II}} = q_1^v$. (ii) If $q_1^v > n - n_c$, then*

$$q_1^{\mathrm{II}} \leq \max\left\{1, \left\lceil \sqrt{\rho_{1,2}(n - n_c)} \right\rceil\right\},$$

*and the welfare loss is*

$$W_{\max} - W(s^{\mathrm{II}}) = (p_1 - p_2)(n - q_1^{\mathrm{II}}).$$

*If $\beta \to \infty$, we have $q_1^{\mathrm{II}} = n$ and $W_{\max} = W(s^{\mathrm{II}})$.*

Theorem A.2 can be proved analogously to the proof of Theorem 4.7.

### A.5. Proof for Theorem 4.9

*Proof.* Based on Lemma A.1, we have $q_1(c) \geq 1$ and $q_j \in \{0, 1\}$ for $j \neq 1$, meaning individual creators select distinct strategies $e_j$. Additionally, $q_1 = q_1(c)$ in this game, where individual creators have restricted action sets. The group utility is defined as follows:

For $1 \leq q_1 \leq K$, the group utility is:

$$Q_c(q_1) = p_1 \frac{q_1 \exp(\beta^{-1})}{q_1 \exp(\beta^{-1}) + (K - q_1)} + p_2(n_c - q_1). \tag{9}$$

Let $b = \exp(\beta^{-1}) - 1 \in (0, \infty)$. We compute the derivative of $Q_c(q_1)$ with respect to $q_1$:

$$Q_c'(q_1) = \frac{n+1}{2n} \cdot \frac{(b+1)K}{(q_1 b + K)^2} - \frac{1}{2n}.$$

By selecting a sufficiently large $\beta$, we ensure $Q_c'(q_1) > 0$ for $1 \leq q_1 \leq K$.

For $q_1 \geq K + 1$, the group utility simplifies to:

$$Q_c(q_1) = p_1 + p_2(n_c - q_1).$$

Thus, when $1 \leq q_1 \leq K$, $Q_c(q_1)$ increases as $q_1$ increases; when $q_1 \geq K + 1$, $Q_c(q_1)$ decreases as $q_1$ increases. Therefore, the optimal value $q_1^\star = K$ maximizes the group utility. Combining this result with Eq. 8, we complete the proof. □

Note that for a more general unbalanced user distribution satisfying $p_1 \geq \frac{1}{n}$, $p_1 + (n-1)p_2 = 1$, and $p_2 = p_3 = \cdots = p_n$, the results still hold when $p_1 > Kp_2$. Moreover, when $p$ is uniform, the value of $K$ no longer affects user welfare in this setting.

### A.6. Restricted MTvN Game

In the Restricted MTvN game, there are $L$ groups, each of size $h$, with user distribution $p_\ell = \frac{n+1}{n(L+1)}$ for $\ell \in [L]$ and $p_{L+1} = \cdots = p_n = \frac{1}{n(L+1)}$. Each group specializes in specific topics: for group $\ell \in [L]$, the action set is restricted to $\{e_\ell\} \cup \{e_{L+1}, e_{L+2}, \ldots, e_n\}$. This setup models scenarios where creators with expertise in specific topics form groups to maximize their collective utility.

**Corollary A.3.** *In a Restricted MTvN game, the following results hold:*

1. *When $r_1 = r_2 = \cdots = r_K = 1$, if $\beta \to 0$, then $W(s^{\mathrm{g}}, K) = 1$. If $\beta \to \infty$, then $W(s^{\mathrm{g}}, K) = W_{\max}$.*

2. *If $r_1 = \cdots = r_\tau = 1$ and $r_{\tau+1} = \cdots = r_n = 0$, and $h > \tau$ then:*

$$W(s^{\mathrm{II}}, K) = \begin{cases} \frac{(n+1)L}{n(L+1)} K + \frac{n - KL}{n(L+1)}, & K \leq \tau, \\ \frac{(n+1)L}{n(L+1)} \tau + \frac{n - KL}{n(L+1)}, & 1 + \tau \leq K \leq h. \end{cases}$$

### A.7. Proof for Theorem 4.12

*Proof.* For each group $\ell \in [L]$, let $q_j(\ell)$ denote the number of creators in group $\ell$ selecting strategy $e_j$. If the group pure Nash equilibrium $\{q_j\}_{j=1}^m$ differs from the vanilla equilibrium $\{q_j^{\mathrm{v}}\}_{j=1}^m$, there exist $j, k \in [m]$ such that $q_j < q_j^v$ and $q_k > q_k^v$. We focus on the group utility for users $j$ and $k$. As $\beta \to 0$, the vanilla equilibrium $\{q_j^{\mathrm{v}}\}_{j=1}^m$ satisfies:

$$\frac{p_j}{q_j^v} \geq \frac{p_k}{q_k^v + 1}, \quad \frac{p_j}{q_j^v + 1} \leq \frac{p_k}{q_k^v}. \tag{10}$$

Next, we compute the reward increment when a creator in group $\ell$ switches from strategy $e_k$ to $e_j$:

$$Q_\ell\left(q_j(\ell)+1, q_k(\ell)-1\right) - Q_\ell\left(q_j(\ell), q_k(\ell)\right) \tag{11}$$

$$= v_j(q_j+1) \cdot (q_j(\ell)+1) - v_j(q_j) \cdot q_j(\ell) + v_k(q_k-1) \cdot (q_k(\ell)-1) - v_k(q_k) \cdot q_k(\ell)$$

$$= \frac{p_j}{q_j+1}(q_j(\ell)+1) - \frac{p_j}{q_j}q_j(\ell) + \frac{p_k}{q_k-1}(q_k(\ell)-1) - \frac{p_k}{q_k}q_k(\ell)$$

$$= \frac{p_j}{q_j+1}\left(1 - \frac{q_j(\ell)}{q_j}\right) - \frac{p_k}{q_k-1}\left(1 - \frac{q_k(\ell)}{q_k}\right).$$

For every group $\ell \in [L]$ with $q_k(\ell) \geq 1$, if $q_j(\ell) = 0$, then based on (10) and the conditions $q_j \leq q_j^v - 1$ and $q_k \geq q_k^v + 1$, we have:

$$Q_\ell(q_j(\ell)+1, q_k(\ell)-1) - Q_\ell(q_j(\ell), q_k(\ell)) > 0.$$

This implies that a creator in this group can increase the group utility by switching from $e_k$ to $e_j$, contradicting the assumption that $\{q_j\}_{j=1}^m$ is a multi-group PNE. Therefore, for every $\ell \in [L]$ with $q_k(\ell) \geq 1$, we have $q_j(\ell) \geq 1$.

Let $C_{j,k}$ denote the set of groups $\ell$ where $q_k(\ell) \geq 1$. Since no group has an incentive to change strategies, for every $\ell \in C_{j,k}$, (11) $\leq 0$. As $q_k(\ell) < q_k$, $|C_{j,k}| \geq 2$. Summing over all $\ell \in C_{j,k}$, we obtain:

$$\sum_{\ell \in C_{j,k}} Q_\ell\left(q_j(\ell)+1, q_k(\ell)-1\right) - Q_\ell\left(q_j(\ell), q_k(\ell)\right) \leq 0.$$

Given that $\sum_{\ell \in C_{j,k}} q_j(\ell) \leq q_j$ and $\sum_{\ell \in C_{j,k}} q_k(\ell) = q_k$, we derive:

$$\frac{p_j}{q_j+1} \leq \frac{p_k}{q_k-1}. \tag{12}$$

Combining the vanilla equilibrium conditions (10) with (12), and noting that $q_j \leq q_j^v - 1$ and $q_k \geq q_k^v + 1$, we have:

$$\frac{p_k}{q_k^v+1} \leq \frac{p_j}{q_j^v} \leq \frac{p_j}{q_j+1} \leq \frac{p_k}{q_k-1} \leq \frac{p_k}{q_k^v}.$$

This implies $q_k - q_k^v \leq 2$. We now consider two cases:

(1) If there exists $k \in [m]$ such that $q_k - q_k^v = 2$, then:

$$\frac{p_k}{q_k^v+1} \leq \frac{p_j}{q_j^v} \leq \frac{p_j}{q_j+1} \leq \frac{p_k}{q_k-1} = \frac{p_k}{q_k^v+1}. \tag{13}$$

All inequalities in (13) must hold as equalities, implying $q_j = q_j^v - 1$.

(2) If for all $k \in [m]$ satisfying $q_k < q_k^v$, $q_k - q_k^v = 1$, combining (10) and (12), we have:

$$\frac{p_j}{q_j+1} \leq \frac{p_k}{q_k-1} = \frac{p_k}{q_k^v} \leq \frac{p_j}{q_j^v - \frac{p_j}{p_k}}.$$

Thus, $q_j \geq q_j^v - \frac{p_j}{p_k}$.

Since $q_j = q_j^v + (q_j^v - q_j)$, there are at least $(q_j^v - q_j)$ strategies $e_k$ with $q_k - q_k^v = 1$, each involving at least 2 groups. Given that $q_j(\ell) \geq 1$ for every $\ell \in [L]$, there are at least $2(q_j^v - q_j)$ distinct groups in $e_j$. Therefore:

$$q_j \geq 2(q_j^v - q_j).$$

This yields $q_j \geq \frac{2}{3}q_j^v$, completing the proof. $\square$

# B. Proofs for Section 5

## B.1. Submodularity of user welfare

**Lemma B.1.** *[Submodularity of Welfare] For any $s = (s_1, \cdots, s_n)$, let $S = \{s_1, \cdots, s_n\}$. Then the social welfare function defined in Eq (2) is submodular as a set function, i.e., for any $S, s_x, s_y$ it holds that*

$$W(S \cup \{s_x\}) - W(S) \geq W(S \cup \{s_x, s_y\}) - W(S \cup \{s_y\}). \tag{14}$$

*Proof.* It is sufficient to prove that for any fixed $K$ and any fixed user $x_j$,

$$\pi_j(\mathcal{T}_j(S \cup \{s_x\})) - \pi_j(\mathcal{T}_j(S)) \geq \pi_j(\mathcal{T}_j(S \cup \{s_x, s_y\})) - \pi_j(\mathcal{T}_j(S \cup \{s_y\})).$$

**Case 1:** If $s_x \notin \mathcal{T}_j(S \cup s_x)$, then LHS $= 0 =$ RHS.

**Case 2:** If $s_x \notin \mathcal{T}_j(S \cup \{s_x, s_y\})$, then LHS $\geq 0$ and RHS $= 0$.

Next we assume that $s_x \in \mathcal{T}_j(S \cup \{s_x, s_y\})$. We denote the relevance scores of $s_x, s_y$ toward this user as $\sigma_x$ and $\sigma_y$.

**Case 3.1:** If $s_y \notin \mathcal{T}_j(S \cup \{s_y\})$, then RHS $=$ LHS.

**Case 3.2:** If $s_y \notin \mathcal{T}_j(S \cup \{s_x, s_y\})$ and $s_y \in \mathcal{T}_j(S \cup \{s_y\})$, so $\sigma^y$ ranks as the last one in $\mathcal{T}_j(S \cup \{s_x, s_y\})$,

we have

$$\text{LHS} - \text{RHS} = r_K(\sigma_{K-1} - \sigma_K) - r_K(\sigma_{K-1} - \sigma^y) = r_K(\sigma^y - \sigma_K) \geq 0.$$

**Case 3.3:** If $s_y \in \mathcal{T}_j(S \cup \{s_x, s_y\})$ and $\sigma^y \geq \sigma^x$, and we suppose that $\sigma^x$ replace the original relevance score in $s_y \in \mathcal{T}_j(S \cup \{s_x\})$ at index $i^x$, then

$$\text{LHS} - \text{RHS} = (r_{i^x} - r_{i^x+1})(\sigma^x - \sigma_{i^x}) + \sum_{t=i^x+1}^{K-1} (r_t - r_{t+1})(\sigma_t - \sigma_{t+1}) + r_K(\sigma_{K-1} - \sigma_K) \geq 0.$$

**Case 3.4** If $s_y \in \mathcal{T}_j(S \cup \{s_x, s_y\})$ and $\sigma^y < \sigma^x$, and we suppose that $\sigma^y$ replace the original relevance score in $s_y \in \mathcal{T}_j(S \cup \{s_x, s_y\})$ at index $i^y$, then

LHS $-$ RHS

$$= r_{i^y}(\sigma_{i^y-1} - \sigma_{i^y}) - r_{i^y}(\sigma_{i^y} - \sigma^y) + r_{i^y+1}(\sigma_{i^y} - \sigma_{i^y+1}) - r_{i^y+1}(\sigma^y - \sigma_{i^y+1}) + \sum_{t=i^y+1}^{K-1} r_{t+1}(\sigma_t - \sigma_{t+1}) - r_{t+1}(\sigma_t - \sigma_{t+1})$$

$$= r_{i^y}(\sigma^y - \sigma_{i^y}) + r_{i^y+1}(\sigma_{i^y} - \sigma_{i^y+1}) - r_{i^y+1}(\sigma^y - \sigma_{i^y}) + \sum_{t=i^y+1}^{K-1} r_{t+1}(\sigma_t - \sigma_{t+2})$$

$$\geq r_{i^y}(\sigma^y - \sigma_{i^y}) - r_{i^y+1}(\sigma^y - \sigma_{i^y}) = (r_{i^y} - r_{i^y+1})(\sigma^y - \sigma_{i^y}) \geq 0.$$

$\square$

We can further extend (14) to a more general version: For any $S, S', S''$ it holds that

$$W(S \cup S'') - W(S) \geq W(S \cup S' \cup S'') - W(S \cup S'). \tag{15}$$

It is sufficient to prove that for any $S, S', s_x, s_y$ it holds that

$$W(S \cup \{s_x, s_y\}) - W(S) \geq W(S \cup S' \cup \{s_x, s_y\}) - W(S \cup S').$$

Based on Lemma B.1, we have

$$\begin{aligned}
\text{LHS} &= W(S \cup \{s_x, s_y\}) - W(S \cup \{s_y\}) + W(S \cup \{s_y\}) - W(S) \\
&\geq W(S \cup S' \cup \{s_x, s_y\}) - W(S \cup S' \cup \{s_y\}) + W(S \cup S' \cup S'\{s_y\}) - W(S \cup S') \\
&= \text{RHS}.
\end{aligned}$$

## B.2. Proof for Theorem 5.4

*Proof.* We set $\widetilde{W}(s) = \beta W(s)$, $\widetilde{\sigma}(s, x) = \beta^{-1}\sigma(s, x)$. Under a rescaling of constant $\beta$, it is without loss of generality to consider a scoring function $\sigma \in \left[0, \frac{1}{\beta}\right]$, the user utility function, and the social welfare function in the following form. $\mathcal{T}_j(S; K)$ represents the strategies $s$ with the top-$K$ relevance scores for user $x_j$, where $S = s_1, \cdots, s_n$. If $|S| < K$, we complement $S$ with strategy $\mathbf{0}$ to ensure $|S| = K$, as this does not affect the result. We prove the theorem for the fixed $K$. We ignore $K$ in $\mathcal{T}_j(S)$, and write as $\mathcal{T}_j(S)$ in abbreviation.

$$Q_i(\boldsymbol{s}_i; \boldsymbol{s}_{-i}) = \sum_{i \in \mathcal{C}_\ell} \sum_{j=1}^m p_j \pi_j(S) \mathbb{I}\left[s_i \in \mathcal{T}_j(S; K)\right] \frac{\exp\left(\sigma(s_i, x_j)\right)}{\sum_{s' \in \mathcal{T}_j(S;K)} \exp\left(\sigma(s', x_j)\right)},$$

$$W(S) = \sum_{j=1}^m \pi_j(S) = \sum_{j=1}^m \sum_{k=1}^K \sigma_{j,l(k)}.$$

We need to prove that

$$Q_\ell(\boldsymbol{s}_\ell, \boldsymbol{s}_{-\ell}) \geq \frac{1}{K}\left[W(S) - W(S_{-\ell})\right]. \tag{16}$$

It is sufficient to prove that for any $j \in [m]$,

$$\beta \pi_j(S) \sum_{i \in \mathcal{C}_\ell} \mathbb{I}\left[s_i \in \mathcal{T}_j(S)\right] \frac{\exp\left(\sigma(s_i, x_j)\right)}{\sum_{s' \in \mathcal{T}_j(S)} \exp\left(\sigma(s', x_j)\right)} \geq \beta \frac{1}{K}\left[\pi_j(S) - \pi_j(S_{-\ell})\right]. \tag{17}$$

When $s_i \notin \mathcal{T}_j(S)$ for any $i \in \mathcal{C}_\ell$, (17) is trivially held because LHS = RHS = 0. With a little abuse of notation, next we assume that there are $h$ content of group $\ell$ in $\mathcal{T}_j(S)$. Next we let

$$\{\exp(\sigma(\boldsymbol{s}, x_j)) \mid \boldsymbol{s} \in \mathcal{T}_j(S)\} = \{v_1, v_2, \ldots, v_h, v_{h+1}, \ldots, v_K\},$$

and

$$\{\exp(\sigma(\boldsymbol{s}, x_j)) \mid \boldsymbol{s} \in \mathcal{T}_j(S_{-\ell})\} = \{v_1', v_2', \ldots, v_h', v_{h+1}, \ldots, v_K\},$$

where $v_i \geq v_i'$ for $i \in [h]$, and $h \leq \min\{K, |\mathcal{C}_\ell|\}$, and $v_i \in [1, e^{\frac{1}{\beta}}]$.

We let

$$S_h = v_1 + \cdots v_h, S_K = v_1 + \cdots v_h + v_{h+1} + \cdots + v_K, S_{K-h} = S_k - S_h$$
$$P_h = v_1 v_2 \cdots v_h > 1, P_h' = v_1' v_2' \cdots v_h', P_K = v_1 v_2 \cdots v_K, P_{K-h} = P_K/P_h.$$

So we can rewrite (17) as

$$\frac{S_h}{S_K} \log P_K \geq \frac{1}{K}\left[\log P_h - \log P_h'\right].$$

Noticed that $P_h' \geq 1$, it is sufficient to show that

$$\frac{S_h}{S_K} \log P_K \geq \frac{1}{K} \log P_h. \tag{18}$$

Let

$$f(v_1, \cdots, v_K) = \frac{S_h \log P_K}{S_K \log P_h}.$$

For $i \in [h+1, K]$,

$$\frac{\partial f}{\partial v_i} = \frac{S_h}{\log P_h} \cdot \frac{\frac{1}{v_i} S_k - \log P_k}{(S_k)^2}.$$

Since $v_j > \log v_i$ for $v_j \in [1, e^{\frac{1}{\beta}}]$, so $S_k > \log P_k$ when $v_i = 1$. So we have the partial derivative of $v_i$ greater than 0 on $v_i = 1$:

$$\frac{\partial f}{\partial v_i}(v_1, \cdots, v_i = 1, \cdots, v_K) = \frac{S_h}{\log P_h} \cdot \frac{S_k - \log P_k}{(S_k)^2} > 0.$$

Since $\frac{1}{v_i} S_k - \log P_k$ is non-increasing in $v_i$, so the value of function $f$ in non-decreasing or increases first and then decreases with variable $v_i$ increases, so only boundary values of $v_i$ will minimize $f$.

Thus when we intend to minimize $f$, $v_i = 1$ or $e^{\frac{1}{\beta}}$ for $i \in [h+1, K]$. Suppose that there are $w$ $v_i$'s equal to $e^{\frac{1}{\beta}}$. Then

$$S_k = K - h - w + w \cdot e^{\frac{1}{\beta}},$$

$$\log P_{k-h} = \frac{w}{\beta}.$$

So

$$f = \frac{S_h}{\log P_h} \frac{\log P_h + \frac{w}{\beta}}{(S_h + K - h - w + w \cdot e^{\frac{1}{\beta}})}.$$

And noticed that for $w \in [0, K - h]$,

$$\frac{\partial f}{\partial w} \leq 0 \text{ or } \frac{\partial f}{\partial w} \geq 0.$$

Therefore, only boundary values of $w$ will minimize $f$. So $w = 0$ or $w = K - h$.

**(i)** When $w = 0$ or $K = h$, then

$$f = \frac{S_h \cdot \log P_h}{(S_h + K - h) \cdot \log P_h} = \frac{S_h}{S_h + K - h} \geq \frac{h}{h + K - h} = \frac{h}{K} \geq \frac{1}{K}.$$

**(ii)** When $w = K - h \geq 1$, then:

$$f = \frac{S_h \left( \log(P_h) + \frac{K-h}{\beta} \right)}{\left[ S_h + (K - h)e^{\frac{1}{\beta}} \right] \log(P_h)}.$$

Simplify it we have

$$f = \frac{S_h}{S_h + (K - h)e^{\frac{1}{\beta}}} + \frac{S_h(K - h)}{\beta \left[ S_h + (K - h)e^{\frac{1}{\beta}} \right] \log(P_h)} > \frac{S_h}{\beta \left[ S_h + (K - h)e^{\frac{1}{\beta}} \right] \log(P_h)}.$$

By GM-AM inequality, we have $P_h \leq (S_h/h)^h$. So

$$\frac{S_h}{\beta \left[ S_h + (K - h)e^{\frac{1}{\beta}} \right] \log(P_h)} \geq \frac{S_h}{\beta h \left[ S_h + (K - h)e^{\frac{1}{\beta}} \right] \log(S_h/h)}.$$

We let $= \frac{S_h}{h}$, so we have

$$\frac{S_h}{\beta h \left[ S_h + (K - h)e^{\frac{1}{\beta}} \right] \log(S_h/h)} = \frac{1}{\beta \left( h + (K - h)e^{\frac{1}{\beta}} t^{-1} \right) \log t} \geq \frac{1}{\beta \left( 1 + (K - 1)e^{\frac{1}{\beta}} t^{-1} \right) \log t}. \tag{19}$$

we define

$$G(t) = \left( 1 + (K - 1)e^{\frac{1}{\beta}} t^{-1} \right) \log t.$$

We only need to prove that $G(t) \leq K$. We first take the derivative of $G(t)$,

$$G'(t) = \frac{t + (K - 1)e^{\frac{1}{\beta}}(1 - \log t)}{t^2}.$$

Define $H(t) = t + (K-1)e^{\frac{1}{\beta}}(1 - \log t)$. Then we calculate the derivative of $H(t)$,

$$H'(t) = 1 - (K-1)e^{\frac{1}{\beta}} \cdot \frac{1}{t} \leq 0.$$

For $H(t)$ and $\beta \geq 1$

$$H(t = 1) = 1 > 0,$$

$$H(t = e^{\frac{1}{\beta}}) = e^{\frac{1}{\beta}} + (K-1)e^{\frac{1}{\beta}}\left(1 - \frac{1}{\beta}\right) > 0.$$

So $H(t) > 0$ for $t \in [1, e^{\frac{1}{\beta}}]$. So $G(t)$ is non-decreasing for $t \in [1, e^{\frac{1}{\beta}}]$, so $G(t)$ is maximized at $t = e^{\frac{1}{\beta}}$.

Therefore

$$G(t) = G_{\max}(t) = G(t = e^{\frac{1}{\beta}}) = K.$$

Let $\boldsymbol{s} = (\boldsymbol{s}_\ell)_{\ell=1}^L$ and $\boldsymbol{s}^* = (\boldsymbol{s}_\ell^*)_{\ell=1}^L$ be two different strategy profiles, where $s_\ell$ represents group $\ell$'s strategy. First, based on Lemma B.1 and (15), for every $\ell \in [L]$ we have

$$W([\boldsymbol{s}_\ell^*, \boldsymbol{s}_{-\ell}]) - W(\boldsymbol{s}_{-\ell}) \geq W([\boldsymbol{s}_1^*, \cdots, \boldsymbol{s}_{\ell-1}^*, \boldsymbol{s}_\ell^*, \boldsymbol{s}]) - W([\boldsymbol{s}_1^*, \cdots, \boldsymbol{s}_{\ell-1}^*, \boldsymbol{s}]).$$

Summing over all group $\ell$ we obtain

$$\sum_{\ell=1}^L (W([\boldsymbol{s}_\ell^*, \boldsymbol{s}_{-\ell}]) - W(\boldsymbol{s}_{-\ell})) \geq \sum_{\ell=1}^L (W([\boldsymbol{s}_1^*, \cdots, \boldsymbol{s}_{\ell-1}^*, \boldsymbol{s}_\ell^*, \boldsymbol{s}]) - W([\boldsymbol{s}_\ell^*, \cdots, \boldsymbol{s}_{\ell-1}^*, \boldsymbol{s}]))$$

$$= W([\boldsymbol{s}^*, \boldsymbol{s}]) - W(\boldsymbol{s})$$

$$\geq W(\boldsymbol{s}^*) - W(\boldsymbol{s}),$$

where the last inequality holds because the top-$K$ relevance scores are elementwise non-decreasing with the addition of more items (strategies), resulting in non-decreasing user welfare. From the Inequlity (16), we have

$$Q_i(\boldsymbol{s}_\ell^*; \boldsymbol{s}_{-\ell}) \geq \frac{1}{K} \cdot \left[ W([\boldsymbol{s}_\ell^*, \boldsymbol{s}_{-\ell}]) - W(\boldsymbol{s}_{-\ell}) \right], \tag{20}$$

And therefore

$$\sum_{i=1}^n Q_i(\boldsymbol{s}_i^*; \boldsymbol{s}_{-i}) \geq \frac{1}{K} \cdot \sum_{i=1}^n \left[ W([\boldsymbol{s}_i^*, \boldsymbol{s}_{-i}]) - W(\boldsymbol{s}_{-i}) \right]$$

$$\geq \frac{1}{K} [W(\boldsymbol{s}^*) - W(\boldsymbol{s})].$$

Then we can take expectation over $\boldsymbol{s} \sim \boldsymbol{\alpha}$ and obtain

$$\sum_{i=1}^n \mathbb{E}_{\boldsymbol{s} \sim \boldsymbol{\alpha}}[Q_i(\boldsymbol{s}_i^*; \boldsymbol{s}_{-i})] \geq \frac{1}{K} [W(\boldsymbol{s}^*) - \mathbb{E}_{\boldsymbol{s} \sim \boldsymbol{\alpha}}[W(\boldsymbol{s})]]. \tag{21}$$

Therefore, combining the definition of GCCE, we have

$$\mathbb{E}_{\boldsymbol{s} \sim \boldsymbol{\alpha}}[W(\boldsymbol{s})] = \mathbb{E}_{\boldsymbol{s} \sim \boldsymbol{\alpha}}[\sum_{\ell=1}^L Q_\ell(\boldsymbol{s})]$$

$$\geq \mathbb{E}_{\boldsymbol{s} \sim \boldsymbol{\alpha}}[\sum_{\ell=1}^L Q_\ell(\boldsymbol{s}_i^*; \boldsymbol{s}_{-i})]$$

$$\geq \frac{1}{K} \cdot \sum_{\ell=1}^L \left[ \mathbb{E}_{\boldsymbol{s} \sim \boldsymbol{\alpha}}[W([\boldsymbol{s}_\ell^*, \boldsymbol{s}_{-\ell}])] - \mathbb{E}_{\boldsymbol{s} \sim \boldsymbol{\alpha}}[W(\boldsymbol{s}_{-\ell})] \right]$$

$$\geq \frac{1}{K} [W(\boldsymbol{s}^*) - \mathbb{E}_{\boldsymbol{s} \sim \boldsymbol{\alpha}}[W(\boldsymbol{s})]].$$

Rearranging terms we obtain

$$PoA(\mathcal{G}) = \frac{\max_{\boldsymbol{s}} W(\boldsymbol{s})}{\min_{\boldsymbol{\alpha} \in GCCE} \mathbb{E}_{\boldsymbol{s} \sim \boldsymbol{\alpha}}[W(\boldsymbol{s})]} \leq 1 + K. \tag{22}$$

$\square$

### B.3. Proof for Theorem 5.5

*Proof.* We construct a game instance of the bandit $\mathcal{C}^3$ game with user engagement rewards. Let $p_1 = \frac{e^{\frac{1}{\beta}} + K - 1}{2e^{\frac{1}{\beta}} + K - 1}$, $p_2 = \frac{e^{\frac{1}{\beta}}}{2e^{\frac{1}{\beta}} + K - 1}$, and $n = K$. A vanilla equilibrium of this game is $\boldsymbol{s}^{\mathrm{v}} = \{\boldsymbol{s}_i = \boldsymbol{e}_2 \text{ for } i \in [n]\}$, since

$$u_i(\boldsymbol{s}^{\mathrm{v}}) = p_2 = p_1 \frac{e^{\frac{1}{\beta}}}{e^{\frac{1}{\beta}} + K - 1} = u_i(\boldsymbol{s}_i = \boldsymbol{e}_1, \boldsymbol{s}^{\mathrm{v}}_{-i}).$$

The welfare at this equilibrium is $W(\boldsymbol{s}^v) = p_2 K$. For the joint strategy $\boldsymbol{s}^\star = \{\boldsymbol{s}_i = \boldsymbol{e}_1\}$, the welfare is $W(\boldsymbol{s}^\star) = K p_1$. Thus, the PoA satisfies

$$\mathrm{PoA}(\mathcal{G}) \geq \frac{W(\boldsymbol{s}^\star)}{W(\boldsymbol{s}^v)} = \frac{p_1}{p_2} = \frac{e^{\frac{1}{\beta}} + K - 1}{e^{\frac{1}{\beta}}}.$$

$\square$

### B.4. Proof for Proposition 5.6

*Proof.* Based on the proof of Theorem 5.4, it suffices to show that for any $j \in [m]$,

$$\pi_j(S) \sum_{i \in \mathcal{C}_\ell} \mathbb{I}[s_i \in \mathcal{T}_j(S)] \frac{\exp\left(\beta^{-1}\sigma(s_i, x_j)\right)}{\sum_{s' \in \mathcal{T}_j(S)} \exp\left(\beta^{-1}\sigma(s', x_j)\right)} \geq \pi_j(S) - \pi_j(S_{-\ell}). \tag{23}$$

As $\beta \to 0$ and $\sigma \in \{0, 1\}$, assume that $r_1 = r_2 = \cdots = r_\tau = 1$ and $r_{\tau+1} = \cdots = r_n = 0$. If $s_i \notin \mathcal{T}_j(S)$ for any $i \in \mathcal{C}_\ell$, (17) holds trivially since LHS = RHS = 0.

Let

$$\{\sigma(\boldsymbol{s}, x_j) \mid \boldsymbol{s} \in \mathcal{T}_j(S)\} = \{\sigma_1, \sigma_2, \ldots, \sigma_h, \sigma_{h+1}, \ldots, \sigma_K\},$$

and

$$\{\sigma(\boldsymbol{s}, x_j) \mid \boldsymbol{s} \in \mathcal{T}_j(S_{-\ell})\} = \{\sigma'_1, \sigma'_2, \ldots, \sigma'_h, \sigma_{h+1}, \ldots, \sigma_K\},$$

where $\sigma_1 = \sigma_2 = \ldots = \sigma_h = 1$, $\sigma'_i \leq$ for $i \in [h]$, and $h \leq \min\{K, |\mathcal{C}_\ell|\}$. Since $\beta \to 0$ and $\sigma \in \{0, 1\}$, we have

$$\sum_{i \in \mathcal{C}_\ell} \mathbb{I}[s_i \in \mathcal{T}_j(S)] \frac{\exp\left(\beta^{-1}\sigma(s_i, x_j)\right)}{\sum_{s' \in \mathcal{T}_j(S)} \exp\left(\beta^{-1}\sigma(s', x_j)\right)} = \frac{h}{x},$$

where $x$ is the number of relevance scores equal to 1 in $\mathcal{T}_j(S)$, and $x = \pi_j(S)$. Let $y = \pi_j(S_{-\ell})$. Noting that $h \geq x - y$, we obtain for the LHS and RHS of (23):

$$\mathrm{LHS} = x \cdot \frac{h}{x} \geq x - y = \mathrm{RHS}.$$

The proof of Proposition 5.6 can then be completed analogously to the proof of Theorem 5.4 in Appendix B.2. $\square$

## C. Supplementary Material for Section 6

### C.1. LBR algorithm (Yao et al., 2024a;c)

**Algorithm 2** (**LBR**) Local Better Response update at time step $t$

1: **Input:** Learning rate $\eta$, the joint strategy profile $\boldsymbol{s}^{(t)} = (\boldsymbol{s}_1^{(t)}, \cdots, \boldsymbol{s}_n^{(t)})$ at the current step $t$.
2: Generate a random direction $\boldsymbol{g}_i \in \mathbb{S}^d$.
3: **if** $u_i(\boldsymbol{s}_i^{(t)} + \eta \boldsymbol{g}_i, \boldsymbol{s}_{-i}^{(t)}) \geq u_i(\boldsymbol{s}^{(t)})$ **then**
4:     $\boldsymbol{s}_i^{(t+\frac{1}{2})} = \boldsymbol{s}_i^{(t)} + \eta \boldsymbol{g}_i$.
5:     Find $\boldsymbol{s}_i^{(t+1)}$ as the projection of $\boldsymbol{s}_i^{(t+\frac{1}{2})}$ in $\mathcal{S}_i$.
6: **else**
7:     $\boldsymbol{s}_i^{(t+1)} = \boldsymbol{s}_i^{(t)}$
8: **end if**

### C.2. Creators' final strategies visualization

Figure 2 presents a visualization of the creators' final strategies under constant attention scores.

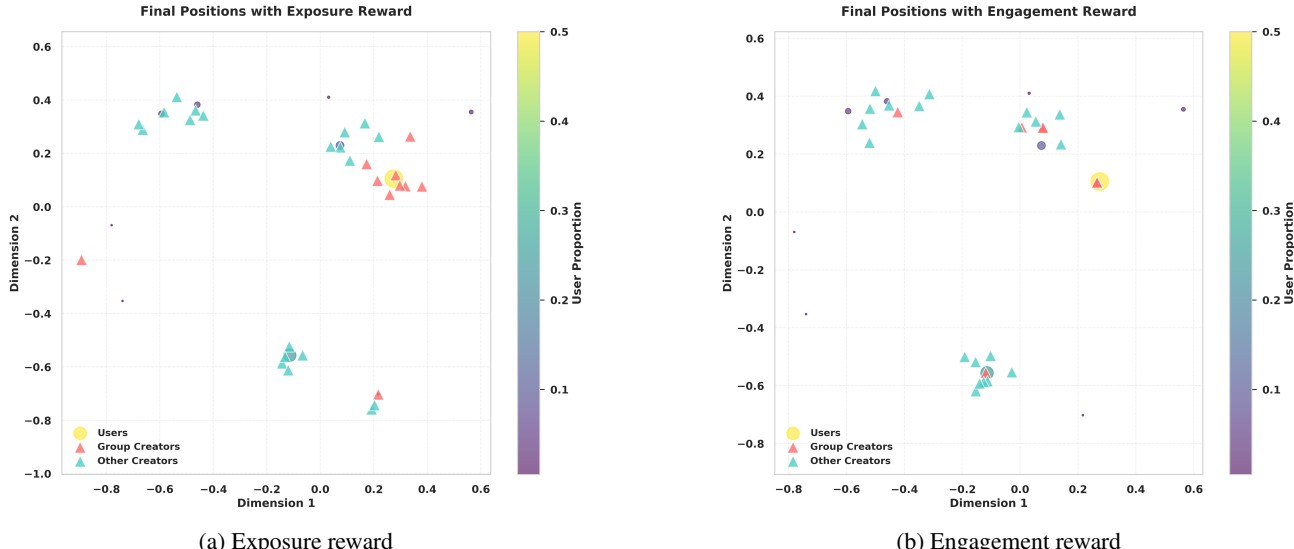

(a) Exposure reward            (b) Engagement reward

Figure 2: Visualization of creators' final strategies under constant attention scores, $n_c = 10$

### C.3. User welfare comparison under different $K$ values

We further validate Theorem 4.9 through simulations with varying values of $K$, using the same user proportions as in the TvN game but with users no longer orthogonal to each other. All group creators are initialized near user $\boldsymbol{x}_1$, while the remaining individual creators are each assigned to a different user $\boldsymbol{x}_j$, where $j \neq 1$. We set $n = 30$, $\beta = 100$, and define the attention scores as $r_1 = r_2 = r_3 = 1$ and $r_i = 0$ for $i > 3$ ($\tau = 3$). The group size is fixed at $n_c = 10$, and the time horizon is set to $T = 1000$. All other parameters follow the simulation setup in Section 6.

We evaluate $K$ over the set $\{1, 2, 3, 6, 10, 15\}$. Results for $K = 1$ and $K = 2$ are not presented graphically. The final user welfare values for $K = 1$ and $K = 2$ are $0.9726 \pm 0.0019$ and $1.9416 \pm 0.0073$, respectively. When $K \leq \tau = 3$, user welfare increases with larger $K$. For $K > \tau$, user welfare slightly declines as $K$ increases. These results align with and further validate the theoretical results of Theorem 4.9.

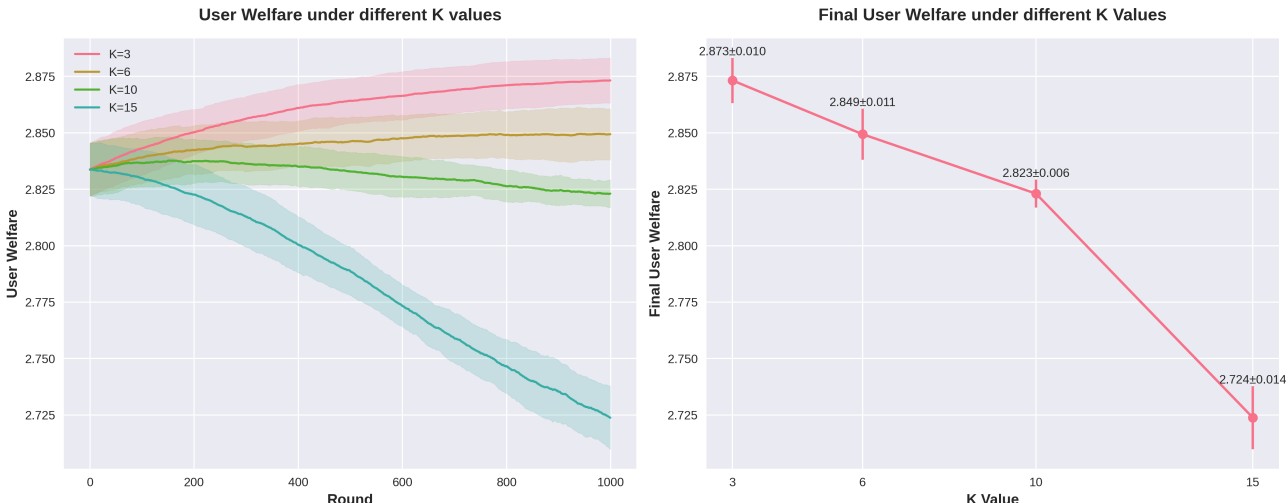

Figure 3: User welfare under different $K$ values

## C.4. User welfare comparison under different $\beta$ values

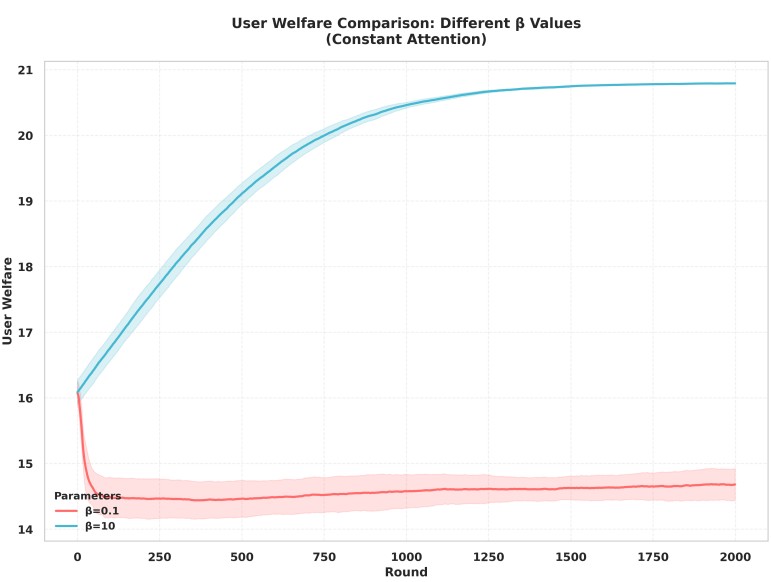

Figure 4: User welfare under different $\beta$ values

We also conduct simulations to validate item 3 of Theorem 4.7. We use the same user proportions as in the TvN game but with users no longer orthogonal to each other. The number of creators is set to $n = 30$, with constant attention scores $r_i = 1$ for all $i \in [n]$, and $K = n$, $T = 2000$. We test two values of $\beta$: $\beta = 0.1$ and $\beta = 10$. All other parameters follow the simulation setup in Section 6. A larger $\beta$ leads to higher user welfare, which aligns with the theoretical findings of item 3 in Theorem 4.7.

## C.5. Additional simulations on Zipf-like user distributions

We further strengthen the practical relevance and credibility of our results by using a more general $\boldsymbol{p}$ based on a Zipf-like or power-law distribution, where $p_j \propto \frac{1}{j^\alpha}$ with $\alpha > 1$. Such distributions are well-documented in real-world platforms and capture the skewed nature of user preferences (Chowdhury & Makaroff, 2013; Cameron, 2022; Chris & Simon, 2022).

Our setup in the main text already follows a similarly skewed pattern. The additional results under this new $p$ yield similar results and insights, further supporting our conclusions. We present results for 3 different values of $\alpha$. In these simulations, we first set $w_j = \max\left\{\frac{1}{j^\alpha}, 0.01\right\}$, and then normalize the user distribution by setting $p_j \propto w_j$.

Table 1: User welfare under different group sizes and reward types for $\alpha = 1.1$, $\alpha = 1.5$, and $\alpha = 1.8$.

| Size | $\alpha = 1.1$ | | $\alpha = 1.5$ | | $\alpha = 1.8$ | |
|---|---|---|---|---|---|---|
| | Exposure | Engagement | Exposure | Engagement | Exposure | Engagement |
| 10 | 4.72±0.04 | 4.81±0.03 | 4.80±0.03 | 4.86±0.01 | 4.85±0.02 | 4.89±0.02 |
| 15 | 4.65±0.05 | 4.79±0.04 | 4.75±0.04 | 4.87±0.02 | 4.84±0.02 | 4.90±0.01 |
| 20 | 4.59±0.07 | 4.81±0.02 | 4.68±0.06 | 4.87±0.02 | 4.73±0.05 | 4.89±0.01 |
| 25 | 4.52±0.08 | 4.82±0.03 | 4.65±0.03 | 4.87±0.02 | 4.70±0.03 | 4.89±0.01 |
| 30 | 3.26±0.02 | 4.77±0.01 | 3.54±0.01 | 4.84±0.02 | 3.75±0.01 | 4.88±0.01 |

### C.6. User welfare comparison under a different creator initialization

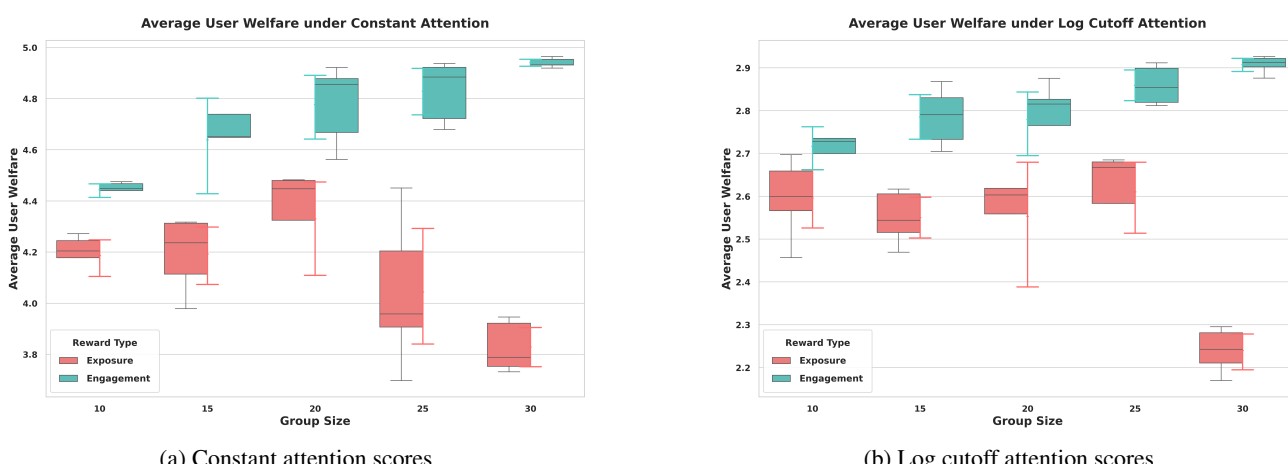

(a) Constant attention scores

(b) Log cutoff attention scores

Figure 5: User welfare under different reward types, group sizes, and attention scores.

We also examine the impact of an alternative creator initialization where all creators are initialized around users 2 to 10. Under this initialization, the resulting user welfare under exposure rewards is worse. The simulation results are presented in Figure 5.

