# OpenReview forum: "Beyond Self-Interest: How Group Strategies Reshape Content Creation in Recommendation Platforms?"
_ICML.cc/2025/Conference — ICML 2025 poster_

### Official Review · Reviewer_SsRd · 2025-02-22

**Overall Recommendation:** 4

**Summary:**

The paper studies group strategies in content creation games: a game framework in which individual creators in a recommender system can form groups. This can lead to interesting scenarios, such as a creator deviating from its strategy in the "vanilla" Nash equilibrium, even though this reduces its individual utility, it increases the group’s utility. They provide game classes, such as the bandit $C^3$ and TvN games, under which the vanilla Nash equilibrium and the $C^3$ game with group creators result in the same or different equilibria. They then analyze general $C^3$ games and the Price of Anarchy under exposure and engagement rewards, showing that it can be unbounded with exposure but bounded for engagement. Finally, they provide simulations supporting their theoretical claims on user welfare.

**Claims And Evidence:**

Yes, the theorems and lemmas are clearly stated and have proofs in the Appendix.

**Essential References Not Discussed:**

Although not an essential reference, I wonder if the authors considered connections to the literature on Algorithmic Collective Action in Machine Learning, https://proceedings.mlr.press/v202/hardt23a.html. Here, too, individuals can form groups to achieve a collective goal.

**Experimental Designs Or Analyses:**

Yes, the experiments are sound and support Theorems 4.7, 4.9, and 5.4. However, some details are missing:
-  Can you specify how gradient descent is used to approximate the optimal group strategy at each round?
-  According to Figure 2 in the appendix, the group creators (red triangles) are dispersed. Just to clarify, this is not for the fixed group strategy as in Section 4.1; here, group creators can have different strategies within a group.

**Methods And Evaluation Criteria:**

Yes, the polarized synthetic dataset and experiments with LBR algorithm support the theoretical claims namely Theorem 4.7, 4.9 and 5.4

**Other Comments Or Suggestions:**

I believe there’s a typo on line 175, column2: “… shapes the same”, i think you mean equal up to permutation of strategies?

**Other Strengths And Weaknesses:**

Overall, I enjoyed reading this paper, and it has clear theoretical contributions and corresponding practical takeaways:

i) The insights from Theorem 4.7 demonstrate how welfare under the single-group Nash equilibrium $s^{II}$ has a drastic decrease compared to individual creators.

ii) The insight from Theorem 4.9, shows how welfare varies with the topK value and that the platform should avoid very large $K$ due to diminishing attention spans.

iii) The Price of Anarchy under exposure and engagement rewards is analyzed and provides a good follow-up to the literature on $C^3$ games PoA from Yao et al., now with group strategies.

The only weakness is in the description of Section 6, and it would improve the paper if the experimental design description were expanded in Section 6 or Appendix C, as I highlighted earlier.

**Questions For Authors:**

1. In the Type 1 and Type 2 equilibria, and more generally for the theorems in Sections 4.1 to 4.4, each individual in the group c plays the same strategy $s_c$, right? This is different from the example in the paragraph on line 172. I believe the results in Theorem 5 do not require this single group strategy.
2. Can you provide more insights on the importance reweighting method mentioned on line 356?

**Relation To Broader Scientific Literature:**

This paper is a good contribution to the literature on content creation games. The concept of producers forming groups is novel and has not been considered in prior work. The technical results are solid and provide a theoretical characterization of new kinds of equilibria, namely: Single-Group Stackelberg Equilibrium (Definition 4.4), Single-Group Nash Equilibrium (Definition 4.5), and scenarios in which they differ from the "no-group" individual creators' Nash equilibrium. The paper also provides Price of Anarchy bounds for general $C^3$ games with group strategies and builds on the PoA characterizations in Yao et al.

**Theoretical Claims:**

Yes, I’ve checked the correctness of Lemma 4.3, Thm 4.6 and 4.7.

A quick clarification on the last statement in section A.1:  For the Bandit C3 game, to get $PNE(G, s_c)$ you initialize all creators in the group $c$ with $s_c$, wlog say these are 1 to $n_c$; now from $n_c$ to $n$ you follow Algorithm 1?

---

> ### Author Rebuttal · Authors · 2025-03-29
>
> >For the Bandit C3 game, to get $PNE(G, s)$ you initialize all creators in the group $c$ with $s_c$, wlog say these are 1 to $n_c$; now from $n_c$ to $n$ you follow Algorithm 1?
>
> We initialize the group of creators with $s_c = (s_1, s_2, \dots, s_{n_c})$, which are not necessarily identical across all members. As you mentioned, once the group strategy $s_c$ is selected, Algorithm 1 is applied to the remaining $n - n_c$ creators, who then sequentially choose their strategies to complete the profile.
>
> >In the Type 1 and Type 2 equilibria, and more generally for the theorems in Sections 4.1 to 4.4, each individual in the group c plays the same strategy $s_c$, right?
>
> Note that $s_c = (s_1, s_2, \dots, s_{n_c})$. Creators within the group are allowed to adopt different strategies (actions), and these strategies are not necessarily identical. In both the Type 1 and Type 2 equilibria, the strategies of group members can differ—the equilibrium does not assume a single shared strategy across all group members. And in Figure 2 of the appendix, the group creators (represented by red triangles) indeed employ different strategies.
>
>
>
> >Can you specify how gradient descent is used to approximate the optimal group strategy at each round?
>
> Thank you for pointing this out. The group aims to select $s_c = (s_1, s_2, \cdots, s_{n_c})$ in order to maximize its group utility, defined as $Q_c(s_c, s_{-c}) = \sum_{i \in \mathcal{C}} u_i(s)$. To approximate the optimal group strategy at each round, the group can perform multiple steps of gradient descent on the objective $Q_c(s_c, s_{-c})$ with respect to the group strategy $s_c$, assuming that the group has full knowledge of the game environment. In more realistic scenarios, the group can instead adopt trial-and-error approaches to iteratively improve its group utility over time. Alternatively, heuristic methods may be employed—for example, assigning certain creators to dominate the exposure of the largest user while others are allocated to target remaining users. This structured, adaptive strategy allows the group to improve utility even with limited information. We will include this clarification in our revised version.
>
>
> >I believe there’s a typo on line 175, column2: “… shapes the same”, I think you mean equal up to permutation of strategies?
>
> Yes, we mean they are equal up to permutation of strategies.
>
> > Can you provide more insights on the importance reweighting method mentioned on line 356?
>
> Thank you for pointing this out. The importance reweighting method, as proposed in (Yao et al., 2024b), enables the platform to steer creator incentives toward under-served users by modifying the reward structure. Specifically, the platform defines the creator's utility as
>
> $$
> u_i(s) = \mathbb{E}_{x \in \mathcal{X}} \left[ w(x) \pi(x, s) P_i(s, x) \right],
> $$
> where $w(x)$ represents the *importance weight* of user $x$. When the platform detects that a user is being under-served under the current content distribution, it increases $w(x)$ for that user. This effectively amplifies the reward for creators who target such users, encouraging them to shift their content in that direction. Over time, this reshapes the content distribution and improves overall user welfare. As a concrete example, consider the game instance in Appendix B.3. In that case, we can set the reward as
>
> $$u_i(s) = w(x_1) \pi(x_1, s) P_i(s, x_1) + w(x_2) \pi(x_2, s) P_i(s, x_2),$$
> and choose $w(x_1)$ to be large and $w(x_2)$ small. This encourages all creators to shift from $s_i = e_2$ to $s_i = e_1$,
> leading to a new PNE where user welfare is optimal. Once this equilibrium is reached, the platform can reset the weights to $w(x_1) = w(x_2) = 1$, and creators will no longer deviate. We will include a detailed explanation and discussion of this method in revision.

---

### Official Review · Reviewer_5tLb · 2025-03-10

**Overall Recommendation:** 3

**Summary:**

This paper investigates group strategic behaviors among content creators in recommendation systems. Specifically, the authors assume that creators within a group can strategically deviate to maximize their collective reward. Using bandit C3 games, they theoretically demonstrate that user welfare can suffer significant losses due to such group strategic behaviors. In more general cases, they analyze the price of anarchy (PoA) of coarse correlated equilibria. Furthermore, they show that user engagement-based reward mechanisms can mitigate these issues compared to exposure-based mechanisms. Simulations support the theoretical findings.

---

I have updated the score to 3 after rebuttal.

**Claims And Evidence:**

**(Pro)** The theoretical results are generally sound.

**(Con)** One major issue is the definition of group deviations. The formation of such groups is questionable, as some players in the group may be worse off due to their inclusion. Consequently, the stability of these groups is unclear, and some players may have an incentive to deviate from the group strategy. A more reasonable setting would introduce an additional requirement ensuring individual rationality—i.e., players should not be worse off by following the group strategy.

**(Con)** Another major issue is the simplification of theoretical results in the bandit C3 game. The theoretical results in Section 4 rely heavily on this specific game structure. For example, when the user population vectors are not orthogonal or the creator strategy set is more general (e.g., continuous), it is unclear how the results would extend. Additionally, in Theorems 4.7 and 4.9, the findings seem highly dependent on the choice of $p_1, p_2, \dots, p_n$. It would be helpful to understand how the results hold for different choices of these parameters.

**Essential References Not Discussed:**

No essential references appear to be missing.

**Experimental Designs Or Analyses:**

**(Con)** Similar to the claims section, I would expect the authors to conduct experiments under more general settings, such as varying the choice of the vector $p$. Additionally, incorporating a real-world dataset with user features would strengthen the analysis.

**Methods And Evaluation Criteria:**

N/A

**Other Comments Or Suggestions:**

1. What is the optimistic tie-breaking rule in Line 203, and how does it relate to the tie-breaking rule in Theorem 4.6?

Some typos were found:
1. The notation $\sigma$ is inconsistent throughout the paper, appearing in three different forms: $\sigma(\cdot, \cdot)$, $\sigma_{\cdot, \cdot}$, and $\sigma_{\cdot}(\cdot)$.
2. Line 172: ${q_j^v}$ and $q_j^{eq}$ are not consistent.
3. Line 214: "for for" → "for"
4. Line 224: One of $s_{1,2,4}$ or $s_{4,5}$ in $S^{II}$ appears to be incorrect.

**Other Strengths And Weaknesses:**

No additional strengths or weaknesses were identified.

**Questions For Authors:**

See the cons listed above.

**Relation To Broader Scientific Literature:**

**(Pro)** The paper's major contribution is the consideration of group strategic behaviors in recommendation systems.

**Theoretical Claims:**

I did not check the details of the theoretical claims, but the results appear to be generally sound.

---

> ### Author Rebuttal · Authors · 2025-03-29
>
> >One major issue is the definition of group deviations.
>
> As we discussed in the paper (Lines 270-274), suppose that all creators have reached an equilibrium, and then some creators decide to form a group. Joining such a group becomes a dominant strategy because the group utility is at least as high as the individual case—at the very least, it remains unchanged if the creators continue their previous actions. If the group generates additional rewards, the bonus can be allocated among the group members, ensuring that each creator receives a reward greater than or equal to their original reward. Given this allocation rule, the formation of such groups is reasonable. Even if a creator deviates from the group strategy, they have an incentive to re-join the group. Furthermore, in the real world, groups of creators often exist and are typically united by a media company, such as an MCN (Lines 28–33).
>
> >When the users are not orthogonal or the creator strategy set is more general, it is unclear how the results would extend.
>
> We use this simplified game to better unveil the theoretical insights. In Section 5, we address the general case, where the user population vectors are not orthogonal, and no assumptions are made regarding the parameter $p$. Theorem 5.3 demonstrates that the PoA under exposure reward can be arbitrarily bad, which aligns with the results in Theorem 4.7. Additionally, we provide bounds on the PoA under engagement rewards in the general case. Furthermore, our empirical results in Section 6 and Appendix C further support the theoretical claims in the general cases, where the user population vectors are not orthogonal and the creator strategy set is continuous.
>
> >In Theorems 4.7 and 4.9, the findings seem highly dependent on the choice of $p$. It would be helpful to understand how the results hold for different choices of these parameters.
>
> Thank you for your valuable suggestions.
> 1) To emphasize the potential negative impact of group strategic behavior on user welfare, we consider the TvN game as an example for worst-case analysis (Yao et al., 2024a). The TvN game, although stylized, reflects user distributions in real-world online content platforms, which are often highly skewed and unbalanced.
> 2) Our choice of this representative $p$ is also primarily for clarity and simplification of presentation. Analogous to the proof of Theorems 4.7 and 4.9, we can extend our results to a more general unbalanced case, where the largest user proportion in the TvN game can vary, and this will provide a more smooth transit from the extreme case to the even case. We will incorporate these results into the revision. And we will expand the discussion to explicitly cover how the results extend to general $p$. In particular:
>     - Under unbalanced $p$, similar welfare loss results hold, and group strategic behavior remains impactful.
>     - Under more evenly distributed $p$, the impact of group behavior diminishes, as also implied by Theorem 4.6 and Theorem 4.12. For example, when $p$ is uniform, in Theorem 4.7, the welfare under both the individual case and the $n_c = n$ case equals 1, and the $K$ will no longer affects the user welfare in Theorem 4.9.
>
>
> >I would expect the authors to conduct experiments under varying choice of the vector $p$.
>
> Thanks for your thoughtful suggestion. We will further strengthen our results by using a more general $p$ based on a Zipf-like or power-law distribution, where $p_j \propto \frac{1}{j^\alpha}$ with $\alpha > 1$. Such distributions have been widely observed in user preferences across online platforms[1][2][3]. Our current $p$ already follows a similarly skewed pattern, and we will clarify this in our revision. The additional experiment results under this new $p$ yield similar results and insights, further supporting the robustness of our conclusions. Please refer to the empirical results provided in our response to reviewer dTVP.
>
>
> [1] Chowdhury et al. Popularity growth patterns of youtube videos-a category-based study.
>
> [2] Cameron, S. Zipf’s Law across social media.
>
> [3] Mosaic Ventures. The creator economy: a power law.
>
> >What is the optimistic tie-breaking rule in Line 203, ...?
>
> Thank you for pointing this out. It refers to selecting $s_{-c}$ in a way that is most favorable to the group. This rule is specific to the equilibrium introduced in Line 203 and is unrelated to the tie-breaking rule mentioned in Theorem 4.6.
> In Theorem 4.6, the tie-breaking rule applies to individual creators when their utilities are equal for selecting different strategies. In such cases, a more general and consistent rule is to assume that creators will choose the user with the smaller index, which is generally used in game theory literature. This deterministic rule avoids ambiguity and is suitable for all cases discussed in the paper. In our revision, we will remove the optimistic tie-breaking rule and adopt this general tie-breaking assumption for clarity and consistency.

---

> > ### Comment · Reviewer_5tLb · 2025-04-02
> >
> > Thank you for the response. I would like to raise my score to 3.

---

### Official Review · Reviewer_vc1Y · 2025-03-12

**Overall Recommendation:** 3

**Summary:**

The paper studies how strategic behaviors of a group of creators can affect user welfare with game-theoretic analysis. In particular, they adopt the content creation competition (CCC) game-theoretic framework introduced in prior works, where users and content creators both derive utilities from the recommender system, with a specific definition of utility for both sides. And they assume there are groups of content creators who collaboratively maximize their group utility.

They conducted game-theoretical analysis under various settings. They found that (1) when the group size is small, groups do not affect the equilibrium much; (2) when group size is large,  user utility suffers a lot. (3) the equilibrium can be quite different under different parameters in the CCC framework, (3) the price of anarchy can be arbitrarily bad when the reward for the creators are exposure (rewarded when created items exposed to users), when the reward for the creators are user engagement( rewarded when created items get user engagement), then the PoA is bounded, suggesting that we might want to use engagement reward to improve user welfare.

**Claims And Evidence:**

I do not find problems

**Essential References Not Discussed:**

I am not aware of any

**Experimental Designs Or Analyses:**

I did not find issues.

**Methods And Evaluation Criteria:**

The paper focuses a bit too much on settings that are not algined with real-world applications.

For instance, I think content creators are rewarded for engagement, but the paper focuses a lot on the setting where creators are rewarded for exposure.

Another example is that users do have limited attention, but the paper focuses a lot on the case where users have infinite attention.

**Other Comments Or Suggestions:**

1. In introduction, it might be good to talk about the findings in this work earlier.

2. Many notations are used without definition. e.g. n, K, \beta in introduction.

3. Acronyms used without explanation, e.g., PNE.

**Other Strengths And Weaknesses:**

Strengths

1. The paper studies a very interesting problem----the impact of strategic behaviors of content creators on user welfare in recommender systems. It might have many real-world implications.

2. The paper provided theoretical analysis on the game-theoretic dynamics of grouped content creators and user welfare under various scenarios.

3. The paper draw conclusions and insights from these theoretical analysis which make sense to me.

Weaknesses

1. I think the paper's analysis focuses too much on unrealistic settings as I mentioned above.

2. The writing can be improved.  See below.

**Questions For Authors:**

No

**Relation To Broader Scientific Literature:**

It adopts prior frameworks on game-theoretic analysis in recommender systems with strategic behaviors. This paper specifically focuses on a new aspect where content creators might form groups to compete strategically.

**Theoretical Claims:**

I did not check the proofs.

---

> ### Author Rebuttal · Authors · 2025-03-29
>
> >The paper focuses a bit too much on settings that are not algined with real-world applications. For instance, I think content creators are rewarded for engagement, but the paper focuses a lot on the setting where creators are rewarded for exposure.
>
> As mentioned in the paper (Lines 144-148), the exposure reward mechanism is widely used in both theoretical and empirical settings (Ben-Porat et al., 2019; Hron et al., 2022; Jagadeesan et al., 2023; Meta, 2022; Savy, 2019), making it an important aspect to study in the context of recommendation systems. Furthermore, both exposure and engagement metrics are used in practice: user engagement tends to be used more often as a reward metric for established creators, while exposure is typically used for new creators (Yao et al., 2023). Thus, our focus on exposure rewards reflects a commonly encountered scenario in many platforms.
>
> >Another example is that users do have limited attention, but the paper focuses a lot on the case where users have infinite attention.
>
> Thank you for your valuable comment. We would like to clarify that our analysis does not assume infinite user attention.
>
> 1) The constant attention model we use does not imply infinite attention. This attention truncated by the parameter $K$ and it assumes users allocate a fixed amount of attention uniformly across the top-$K$ items they are shown—e.g., $r_1 = r_2 = \dots = r_K = 1$. This setting is relevant in practice, particularly in user interfaces where content is displayed in fixed-sized, unordered blocks (e.g., a “For You” page with $K$ equally weighted items). Thus, the model captures a common real-world recommendation scenario without assuming that users consider all available content.
>
> 2) Our paper also considers another setting with *diminishing attention*, modeled as $r_1 = \dots = r_\tau = 1$, $r_{\tau+1} = \dots = r_n = 0$, which captures the case where users only pay attention to the top $\tau$ items. These two types—constant and diminishing—represent common user behaviors corresponding to slow-decay and rapid drop-off attention curves, respectively.
>
> 3) The theoretical results in Section 4 can be extended to more general attention profiles $\{r_i\}_{i=1}^n$. In particular:
>    - Slow decay attention will lead to results akin to those in Theorem 4.7, implying that large group strategic behavior negatively impacts user welfare.
>    - Under rapid drop-off attention, Theorem 4.9 and Corollary 4.10 hold similar results demonstrating that tuning $K$ and $\beta$ can help mitigate user welfare loss.
>
> 4) In our simulations, we also test *log cutoff attention scores* as an intermediate setting. These results are consistent with our theoretical findings, reinforcing the robustness of the insights under various attention models.
>
> We chose to present constant and diminishing attention as representative cases to improve the readability and clarity of the exposition. We will clarify this modeling choice and its implications in our revision.

---

### Official Review · Reviewer_dTVP · 2025-03-14

**Overall Recommendation:** 3

**Summary:**

The paper is attempting to answer : How do group strategies among content creators impact recommendation systems, specifically focusing on content distribution and user welfare? The paper examines how content creator groups impact recommendation systems, contrasting with individual creator behavior. Particularly, they show that  large groups can significantly harm user welfare, especially with exposure-based rewards.

Furthermore, the authors quantifies inefficiency, showing the price of anarchy (PoA) can be arbitrarily large with exposure rewards but is bounded with engagement rewards. They argue and demonstrate that engagement-based rewards better mitigate negative group effects and improve user welfare.

Empirical results from simulations further support the effectiveness of the user engagement rewarding mechanism.

**Claims And Evidence:**

The paper provides theoretical arguments and examples (like the TvN game) to show that group behavior can significantly alter content distribution and user welfare, especially with exposure rewards. This is supported by mathematical formulations and the concept of group equilibria.

The argument that engagement rewards are better for user welfare is supported by the PoA analysis and the simulation results, which show higher user welfare under engagement rewards.

**Essential References Not Discussed:**

The author discussed related work mostly in the study of game-theoretic aspects in recommendation systems. Probably it would be interesting to touch the prior work on empirical studies of creator behavior on content platforms.

**Experimental Designs Or Analyses:**

The experimental designs and analyses are generally sound and appropriate.  The simplifications and assumptions are acknowledged but could be discussed more thoroughly.

**Methods And Evaluation Criteria:**

The game-theoretic framework provides a solid foundation for analysis, PoA offers a quantitative measure of inefficiency, and simulations validate the theoretical findings. The focus on user welfare and the comparison of reward mechanisms are central to the research questions. Therefore, the proposed methods and evaluation criteria make sense for the problem at hand.

**Other Comments Or Suggestions:**

N/A

**Other Strengths And Weaknesses:**

Strengths:

* The paper is generally well-structured with examples that help to clarify the concepts and arguments.
* The combination of theoretical results and simulations strengthens the paper's claims.
* The paper clearly demonstrates the advantages of engagement rewards in mitigating negative impacts of group behavior.
* The paper makes a significant contribution by shifting the focus from individual creator strategies to group strategies in recommendation systems.

Weakness:
* The paper primarily uses synthetic data for simulations. Including empirical validation with real-world data from online platforms would further strengthen the claims and demonstrate the practical relevance of the findings.
* The model relies on certain simplifications and assumptions (e.g., relevance function, user attention scores, specific game setups). The paper could discuss the limitations of these assumptions and their potential impact on the results.

**Questions For Authors:**

In the simulations, were the differences in user welfare between exposure and engagement rewards statistically significant?

**Relation To Broader Scientific Literature:**

The TvN game was introduced by Yao et al. (2024a) to model the dilemma faced by creators in choosing between popular trends and niche topics.  The paper uses the TvN game to illustrate the specific impact of group behavior on content distribution and user welfare. It shows how groups can lead to a significant deviation from the individual creator case, especially with exposure rewards. This provides a concrete example of the general theoretical findings.

**Theoretical Claims:**

I briefly looked at the formulations in the main paper. Do not see any obvious issues.

---

> ### Author Rebuttal · Authors · 2025-03-29
>
> >The paper primarily uses synthetic data for simulations. Including empirical validation with real-world data from online platforms would further strengthen the claims and demonstrate the practical relevance of the findings.
>
> Thank you for the thoughtful suggestion. We will further strengthen the practical relevance and credibility of our results by using a more general $p$ based on a Zipf-like or power-law distribution, where $p_j \propto \frac{1}{j^\alpha}$ with $\alpha > 1$. Such distributions are well-documented in real-world platforms and capture the skewed nature of user preferences [1][2][3]. Our current setup already follows a similarly skewed pattern, and we will clarify this in the revision. The additional results under this new $p$ yield similar results and insights, further supporting our conclusions. We present results for two different $\alpha$ values below.
>
> [1] Chowdhury et al. Popularity growth patterns of youtube videos-a category-based study.
>
> [2] Cameron, S. Zipf’s Law across social media.
>
> [3] Mosaic Ventures. The creator economy: a power law. https://www.mosaicventures.com/patterns/the-creator-economy-a-power-law.
>
> $\alpha=1.1$
> |Group Size|Exp|Eng|
> |-:|:-:|:-:|
> |10|4.72±0.04|4.81±0.03|
> |15|4.65±0.05|4.79±0.04|
> |20|4.59±0.07|4.81±0.02|
> |25|4.52±0.08|4.82±0.03|
> |30|3.26±0.02|4.77±0.01|
>
> $\alpha=1.8$
> |Group Size|Exp|Eng|
> |-:|:-:|:-:|
> |10|4.85±0.02|4.89±0.02|
> |15|4.84±0.02|4.90±0.01|
> |20|4.73±0.05|4.89±0.01|
> |25|4.70±0.03|4.89±0.01|
> |30|3.75±0.01|4.88±0.01|
>
>
>
> >The model relies on certain simplifications and assumptions (e.g., relevance function, user attention scores, specific game setups). The paper could discuss the limitations of these assumptions and their potential impact on the results.
>
> We appreciate your valuable suggestion.
> Our model does make several stylized assumptions as many works in this line also do (Jagadeesan et al., 2023, Hu et al., 2023, Yao et al. 2024a;b). However, we do not see it as a major weakness but rather a necessary simplification to better unveil the theoretical insights. Below, we clarify the rationale and limitations associated with our assumptions:
> 1. **Relevance function**:
> The results in Section 4 are based on the dot product relevance score, which is for the simplification of presentation, and our results also hold for other reasonable relevance functions (e.g., $\sigma$ depending on $||s - x||$). And our general results—Theorems 5.3, 5.4, and 5.5—do not rely on specific assumptions about the form of the relevance function.
> 2. **User attention scores**:
> These two types—constant and diminishing—represent common user behaviors corresponding to slow-decay and rapid drop-off attention, respectively, providing results and insights under these two kinds of user attention.
> 3. **Specific game setups**:
> The bandits $C^3$ game simulates the fundamental scenario where every creator must select a topic to create content. And, the TvN game, where the user distribution is unbalanced, is recognized as a good example for worst-case analysis (Yao et al., 2024a). While generalizing Section 4's results to broader environments is a meaningful and challenging direction for future work, we highlight that Section 5 already addresses the general case: it makes no orthogonality assumptions about user population vectors and does not constrain the parameter $p$.
>
> We will include a dedicated discussion about the limitations and potential impact of these assumptions in our revision.
>
> >In the simulations, were the differences in user welfare between exposure and engagement rewards statistically significant?
>
> Yes, as shown in Figure 1, engagement reward consistently maintain higher user welfare, while exposure rewards lead to a notable decline. Each experiment is run 5 trials to avoid the randomness. We also emphasize that our simulations do not consider the worst-case group behavior under exposure reward. Under such behavior the user welfare could be even lower than reported. Under an alternative initialization where all creators are intialized around users 2–10, the resulting user welfare under exposure reward is worse. We present the empirical results here:
>
> |Group Size|Exp|Eng|
> |:-:|:-:|:-:|
> |10|4.20±0.07|4.53±0.21|
> |15|4.19±0.12|4.54±0.10|
> |20|4.25±0.17|4.69±0.03|
> |25|4.13±0.22|4.81±0.10|
> |30|3.80±0.04|4.95±0.01|
>
> Furthermore, it is important to note that even small improvements in user welfare can have meaningful consequences in practice. For example, platforms like TikTok serve billions of content impressions daily. Thus, marginal gains in user welfare—achieved through better reward design—can translate into substantial improvements in user satisfaction, engagement, and platform revenue.

---

> > ### Comment · Reviewer_dTVP · 2025-04-04
> >
> > I will keep my rating. Thanks for the response

---

### Decision · Program_Chairs · 2025-05-01

**Decision:**

Accept (poster)

**Comment:**

The paper deals with strategic content providers, raising the novel aspect of group strategies. The theoretical contributions are interesting and non-trivial, and the experimental validation helps to demonstrate further how coalition formation shapes content creation.  While the reviewers had their concerns (e.g., assumptions, synthetic experiments), the authors' rebuttal seems to have cleared them all.

I recommend accepting this paper and hope it will impact the "strategic content providers" line of work.